# Modulating gene regulation function by chemically controlled transcription factor clustering

Jiegen Wu [1,2,3,4,5], Baoqiang Chen[4,5], Yadi Liu[1,2,3], Liang Ma[1,2,3], Wen Huang [1,2,3] & Yihan Lin [1,2,3✉]

Recent studies have suggested that transcriptional protein condensates (or clusters) may play key roles in gene regulation and cell fate determination. However, it remains largely unclear how the gene regulation function is quantitatively tuned by transcription factor (TF) clustering and whether TF clustering may confer emergent behaviors as in cell fate control systems. Here, to address this, we construct synthetic TFs whose clustering behavior can be chemically controlled. Through single-parameter tuning of the system (i.e., TF clustering propensity), we provide lines of evidence supporting the direct transcriptional activation and amplification of target genes by TF clustering. Single-gene imaging suggests that such amplification results from the modulation of transcriptional dynamics. Importantly, TF clustering propensity modulates the gene regulation function by significantly tuning the effective TF binding affinity and to a lesser extent the ultrasensitivity, contributing to bimodality and sustained response behavior that are reminiscent of canonical cell fate control systems. Collectively, these results demonstrate that TF clustering can modulate the gene regulation function to enable emergent behaviors, and highlight the potential applications of chemically controlled protein clustering.

[1] Center for Quantitative Biology, Academy for Advanced Interdisciplinary Studies, Peking University, 100871 Beijing, China. [2] The MOE Key Laboratory of Cell Proliferation and Differentiation, School of Life Sciences, Peking University, 100871 Beijing, China. [3] Peking-Tsinghua Center for Life Sciences, Academy for Advanced Interdisciplinary Studies, Peking University, 100871 Beijing, China. [4] School of Life Sciences, Tsinghua University, 100084 Beijing, China. [5]These authors contributed equally: Jiegen Wu, Baoqiang Chen. ✉email: yihan.lin@pku.edu.cn

Recent studies in several areas of biology have suggested that biomolecular condensates may play roles in various cellular functions[1,2], including transcriptional regulation[3–6]. Condensates of transcription factors (TFs) have been observed in multiple gene regulatory systems[7–16]. Mechanistically, liquid-liquid phase separation[8,13,15,16] or alternative mechanisms[17–19] may underlie the formation of TF condensates. At the functional level, these condensate-forming regulators often play key roles in animal cell fate determination[20,21]. Examples include EWS-FLI1[7–9] that drives tumor transformation in Ewing's sarcoma, Bicoid[10] and Zelda[11,12] that participate in *Drosophila* embryogenesis, Oct4[13] that regulates embryonic stem cell pluripotency, GATA3[15] that participates in immune cell fate control, and HOXD13 and RUNX2[16] that play key roles in tissue morphogenesis.

To understand how condensate-forming TFs may control cell fate, it would be helpful to first compare them with canonical cell fate control systems. The canonical systems typically consist of biological circuits, in which individual components interact with one another and form feedback loops. Examples include the differentiation circuit in *Bacillus subtilis*[22], the circuit mediating *Xenopus* oocyte maturation[23], and the lineage-specification circuit during reprogramming[24]. Because of the feedback interactions in the circuits, these systems can display highly nonlinear behaviors[25], including bistability (i.e., the co-existence of two stable states), ultrasensitivity (i.e., increased sensitivity to input change compared to hyperbolic Michaelian response), and memory (i.e., sustained output in response to a transient input). Such emergent functions allow switch-like, all-or-none, and sometimes irreversible transitions between cell states[25]. These properties confer robustness for the control system to perturbations and ensure proper cell fate decisions.

In comparison, while the gene regulatory roles of condensate-forming TFs have been recently characterized in both natural and synthetic systems[13,26–28], we still know little regarding whether such TFs could confer emergent behaviors analogous to canonical cell fate control systems without apparent feedback interactions. Because the condensation process mediated by multivalent interactions is highly cooperative, it has been postulated that the formation of TF clusters may confer ultrasensitivity in gene regulation[3,29]. Consistent with this, our recent work has shown that the co-condensation between TF and coactivator p300 can lead to nonlinear dose–response curve, giving rise to an ultrasensitive switch[28]. Yet it remains to be further determined whether and how clustering (or condensation) represents a general control parameter for the quantitative modulation of the gene regulation function, i.e., the functional dependency of target gene output on the concentration of the TF[30–32] (or input–output relationship).

Due to the complexity of eukaryotic gene regulation[33–37], it is often challenging to clearly delineate the regulatory contribution from a specific parameter of interest, such as the clustering propensity of the TF. More specifically, it is challenging to determine whether the changes in the target gene's response are caused by the clustering of TF, since it is possible that the clustering of TF could be a "passenger" phenomenon accompanying the changes in the target gene's response. Fortunately, synthetic biology offers unique opportunities for tackling such challenges. Synthetic gene regulatory systems can be modularly designed to enable systematic and quantitative interrogations of individual gene regulatory components, and can be designed to behave relatively orthogonally with respect to the endogenous regulation[38–44]. Continuing efforts in synthetic biology have successfully yielded insights into gene regulation[45–50]. Meanwhile, synthetic parts, modules, and systems have been rationally designed for enhancing cellular gene regulatory capacity[51–56], further manifesting the importance of synthetic

biology approaches for understanding and manipulating gene regulatory systems.

Here, we delineated the roles of TF clustering in gene regulation by combining synthetic biology approaches with quantitative single-cell tools. With a synthetic chemically-inducible clustering system, we first provided evidence supporting the casual role of gene expression amplification by the clustering of TFs, and systematically characterized how TF clustering propensity modulates the target gene's transcriptional dynamics. We further showed that the gene regulation function is quantitatively modulated by the TF clustering propensity, with both the effective TF binding affinity and the ultrasensitivity generally increase as the clustering propensity enhances. Intriguingly, under high clustering propensity of the TF, we found that the target gene's response displays bimodality as well as sustained response behavior. These results demonstrate that TF clustering can enable emergent behaviors that are commonly observed in canonical cell fate control systems without apparent feedback loops, and suggest the potential applications of synthetic protein clustering systems for cell state perturbation and control.

## Results

**A bottom-up synthetic biology approach for delineating the role of TF clustering.** To delineate how the clustering of TF molecules influences target gene expression, we needed a TF that could allow tuning its clustering behavior and characterizing the associated changes in its regulatory activity. To achieve such single-parameter tuning of the system, we built upon an existing rapamycin-inducible clustering system[57], and re-designed it to enable titrating the clustering behavior of synthetic TFs (e.g., rTetR-3×VP16) fused with EGFP, with the help of the "clustering mediator" that is also fused with EGFP (Fig. 1a). Without rapamycin, the TF molecules would form tetramers due to the homo-oligomeric tag 6 or HOTag6. In the presence of rapamycin, the TF molecules and the clustering mediator molecules would tend to assemble into clusters because of the rapamycin-mediated interaction between the FRB and FKBP domains (Fig. 1b), and the cluster size would be tuned by the rapamycin concentration.

To quantify the regulatory activity of the synthetic TFs, we constructed a cognate reporter gene that reports the regulator's activity at the level of the target gene's nascent transcriptional activity, as well as at the level of the target gene's protein expression level. More specifically, the reporter gene contains TF binding sites (e.g., TetO) upstream of the miniCMV promoter, driving the expression of the infrared fluorescent protein iRFP together with a 24× PP7 stem-loop cassette[58] (Fig. 1a). When the reporter gene is activated, the iRFP protein would be expressed, and at the meantime, the PP7 cassette would be transcribed and folded into stem loops to recruit cognate RNA binding protein PCP labeled with mCherry (PCP-3×mCherry), allowing real-time quantifications of nascent transcriptional activity of the reporter gene.

**Rapamycin-inducible system allows tuning TF clustering propensity.** We first investigated how rapamycin concentration affects the clustering behavior of TF molecules. Using a U2OS cell line stably integrated with a synthetic TF (Fig. 1a), we titrated cells with a gradient of rapamycin concentration, ranging from 0 nM to 1000 nM, and imaged them under a confocal microscope at multiple z slices. As expected, cells under high rapamycin concentrations often contained large fluorescent foci, representing the clustering of the TF molecules (Fig. 1c). We then detected the clusters with an algorithm (Supplementary Fig. 1a, "Methods"), and quantified the fraction of cells containing detectable clusters and the characteristics (size and number per cell) of the clusters

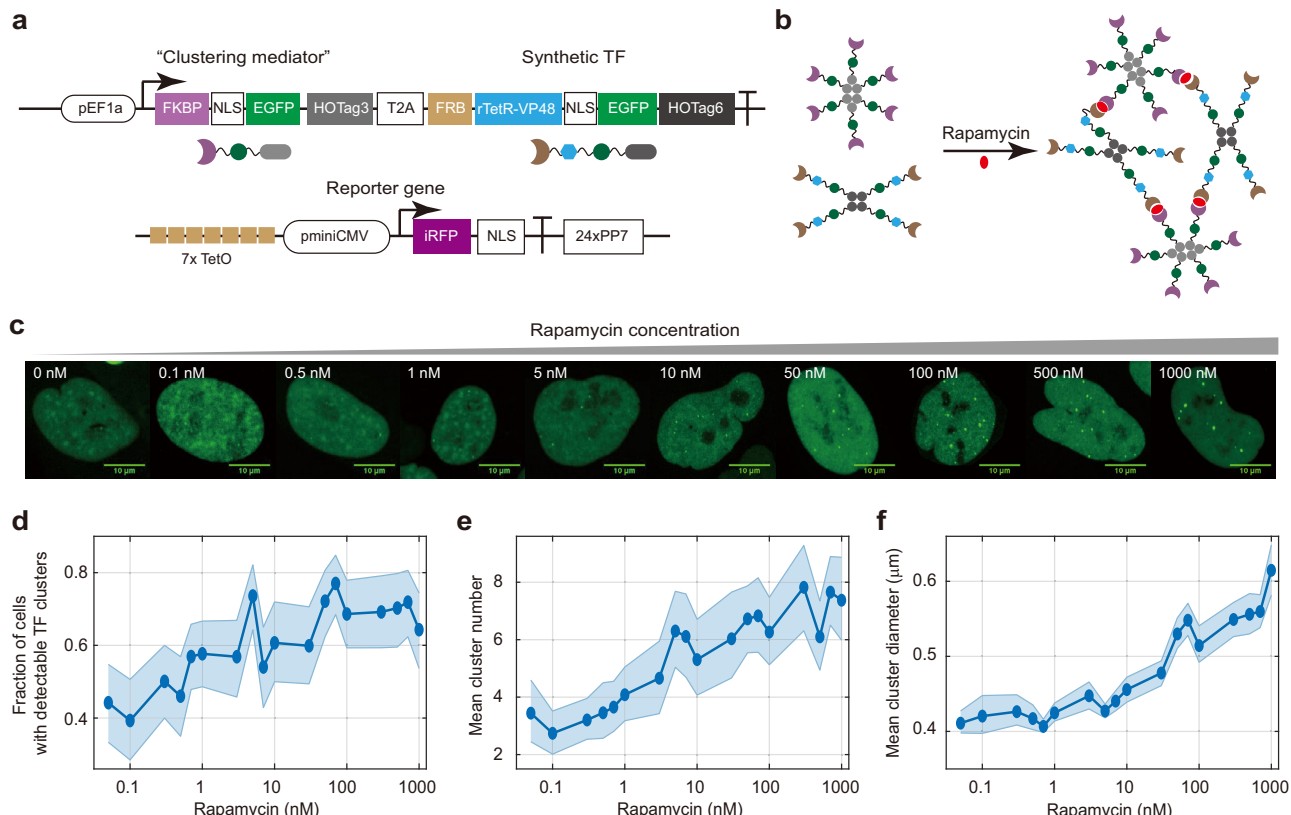

**Fig. 1 A synthetic gene regulatory system with chemically controlled TF clustering propensity. a, b** Schematic of the system design. Our design is built upon a previously reported rapamycin-inducible clustering mechanism. The synthetic TF contains DNA binding domain (e.g., rTetR) and trans-activation domain (e.g., VP48), fused with FRB, EGFP, and a homo-oligomeric tag HOTag6 (**a**). This synthetic TF is co-translated with a "clustering mediator" composed of FKBP, EGFP, and HOTag3. The cognate reporter of the TF contains iRFP and 24x PP7 stem loops. The system also contains a plasmid encoding PCP-3xmCherry for visualizing nascent transcription (not shown). TF clustering is mediated by rapamycin-mediated binding between FRB and FKBP (**b**). **c–f** Rapamycin modulates TF clustering propensity. Representative confocal images (maximum projection of z slices) of monoclonal U2OS cells containing the preceding system were treated with rapamycin at indicated concentrations (**c**). See "Methods" for details on imaging. Scale bars indicate 10 µm. The fraction of cells containing detectable TF clusters (**d**), the averaged detected cluster number per cell (**e**), and the averaged cluster diameter (**f**) are quantified. The rapamycin concentrations are 0.05, 0.1, 0.3, 0.5, 0.7, 1, 3, 5, 7, 10, 30, 50, 70, 100, 300, 500, 700 and 1000 nM (left to right), and the cell numbers are between 70 and 117 (see source data for the exact cell number). The shaded regions indicate 95% CI by bootstrap. See also Supplementary Fig. 1b, c. Source data are provided.

(Supplementary Fig. 1b-c). We found that increasing rapamycin concentration increases the fraction of cluster-containing cells, the mean cluster diameter, and the mean number of clusters per cell (Fig. 1d–f). Interestingly, the curve for mean cluster diameter has not yet reached an apparent saturation at the maximum rapamycin concentration, while other two curves appear to have saturated at high rapamycin concentrations, indicating that TF clusters tend to grow in size instead of in number beyond a threshold rapamycin concentration at ~100 nM (Fig. 1d–f). Because rapamycin alters the likelihood of forming visible clusters in a cell as well as the mean size and number of clusters, we think it is more accurate to describe that rapamycin modulates the synthetic TF's propensity to cluster, compared to other descriptions such as the modulation of cluster size or number.

While these characterizations provide a picture of how rapamycin tunes TF clustering propensity at the level of the cell population, there are additional parameters such as protein expression level that can affect the clustering propensity at the level of individual cells (Supplementary Fig. 1b, c). Note that we observed heterogeneous clustering behaviors of cells under the same rapamycin condition, which could arise from variabilities in TF expression level and/or other cell states. Furthermore, it is likely that our imaging condition cannot capture all functional TF clusters.

To ensure that the formation of clusters was indeed due to rapamycin-induced binding FKBP and FRB, and to delineate the potential reasons for why TF clusters even existed in the absence of rapamycin (Supplementary Fig. 1a, b), we built a two-color construct where the synthetic TF and the clustering mediator are separately fused with EGFP and mCherry (Supplementary Fig. 1d top). We then transfected this construct into U2OS cells, and found that EGFP and mCherry can be induced to form co-localized clusters by rapamycin, consistent with the expected behavior of the system (Supplementary Fig. 1d bottom right). In the absence of rapamycin, we found that each of the two components can separately form cluster-like fluorescence signals, and that these clusters are non-overlapping between the two colors (Supplementary Fig. 1d bottom left). Thus, the observed TF clusters in the absence of rapamycin in the single-color cell line (Supplementary Fig. 1a) likely resulted from the clustering of individual components instead of the co-clustering of the two components.

**TF clustering propensity directly affects target protein expression level**. While TF clustering has been previously suggested to amplify target gene expression inside cells[26–28], additional evidence would

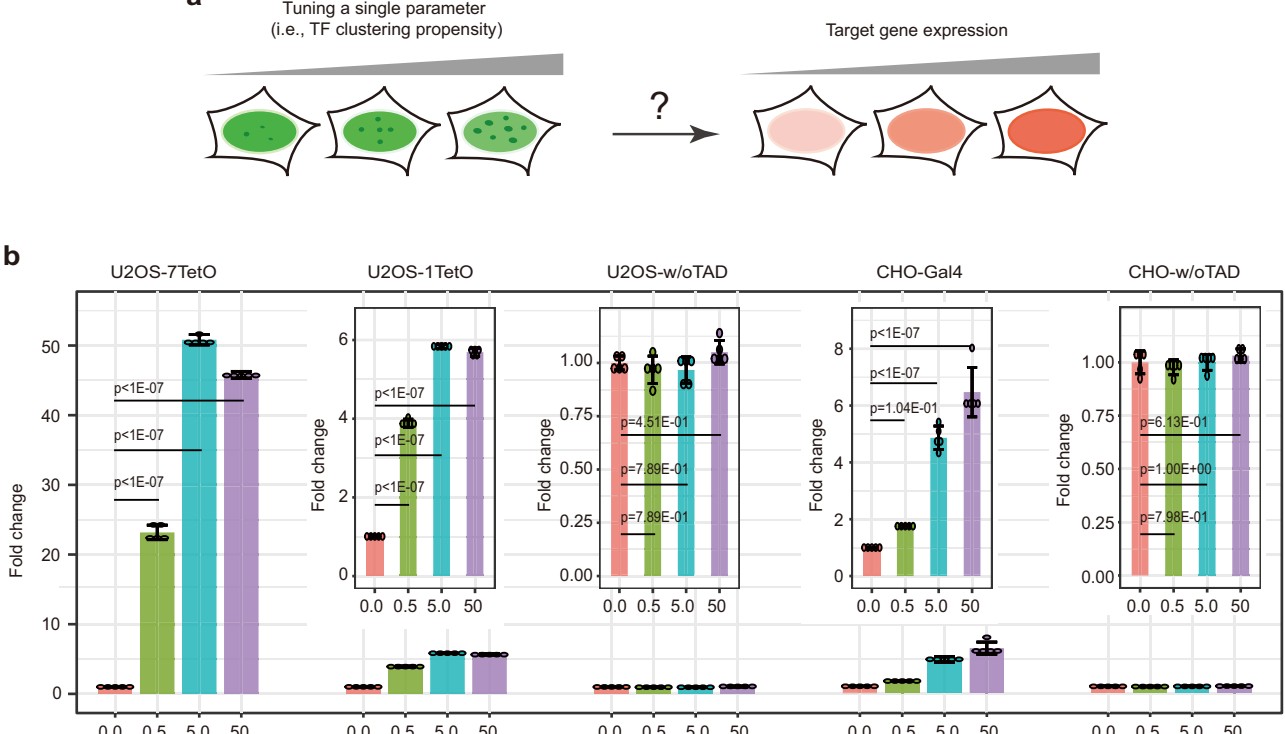

**Fig. 2 TF clustering propensity modulates gene expression amplification. a** Schematic of the experiment. **b** Flow cytometry quantifications of the reporter response under various TF clustering propensity conditions for five separate synthetic systems. In all conditions, iRFP fluorescence levels were quantified 72 h post the addition of doxycycline (0.5 μg/ml for U2OS cells and 0.1 μg/ml for CHO cells) and indicated rapamycin concentration. Fold change was calculated by normalizing to zero rapamycin condition. U2OS-7TetO: Monoclonal U2OS cells containing the system depicted in Fig. 1a; U2OS-1TetO: Analogous to U2OS-7TetO except that the reporter gene contains only 1x TetO site; U2OS-w/oTAD: Polyclonal U2OS cells containing the system depicted in Fig. 1a except that the rTetR is not fused to a trans-activation domain; CHO-Gal4: Monoclonal CHO cells containing the system depicted in Supplementary Fig. 1e; CHO-w/oTAD: Polyclonal CHO cells containing the system depicted in Supplementary Fig. 1e except that the Gal4 is not fused to a trans-activation domain. Data are presented as mean ± S.D. ($n = 5$ biological replicates). $p$ values were calculated by one-way ANOVA and TukeyHSD. Source data are provided, which contain details of statistical tests.

be necessary to further support the causality. To address this, we sought to leverage the single-parameter tuning capability of our synthetic system.

Having established the relationship between clustering propensity and rapamycin concentration, we next characterized the modulation of protein expression level of the target gene (iRFP) by TF clustering propensity (Fig. 2a). Besides TF clustering propensity, we also sought to study the influences of additional parameters of the system, including TF binding site number, DNA binding domain, and host cell line. To do so, we thus used three different cell lines, i.e., the preceding cell line that contains 7xTetO sites in front of the reporter (U2OS-7TetO), another U2OS cell line that contains 1xTetO site in front of the reporter (U2OS-1TetO), and a CHO cell line that contains Gal4-VP64 as the TF (CHO-Gal4, Supplementary Fig. 1e).

With all these three cell lines, we observed that rapamycin significantly affected the target protein expression level (Fig. 2b). Three important control experiments were performed. First, with U2OS-7TetO cells, adding rapamycin alone (without adding doxycycline) did not affect the fluorescence signal (Supplementary Fig. 1f; see Supplementary Fig. 1g for flow cytometry gating strategy), indicating that our system tolerates high rapamycin concentrations, and the observed change in protein expression (Fig. 2b) could not be caused by the formation of TF clusters alone (i.e., it requires DNA binding capability). Second, we constructed a control U2OS cell line (U2OS-w/oTAD) with a

synthetic TF lacking the trans-activation domain (i.e., rTetR only), and observed no modulation of gene expression by rapamycin (Fig. 2b third panel). Third, a similar result was observed for an analogous control CHO cell line (CHO-w/oTAD, Fig. 2b last panel). The latter two experiments suggested that the observed change in target protein expression (Fig. 2b) was not contributed by other components in the system (e.g., EGFP, FKBP, HOTag, etc.).

These preceding data contain several notable quantitative features. First, the extent of expression amplification by TF clustering depends on the DNA binding site number (Fig. 2b, compare U2OS-7TetO with U2OS-1TetO), consistent with results from recent studies[59–61]. Second, the expression amplification can occur at a low rapamycin concentration (i.e., 0.5 nM, Fig. 2b first two panels), where no increase in TF clustering can be visually detected compared to the no rapamycin condition (Supplementary Fig. 1b, c). This result indicates that TF clustering likely occurs at a relatively low rapamycin concentration, but our imaging condition cannot capture small clusters. In other words, our reporter protein (i.e., iRFP) can read out TF clustering in a much more sensitive manner compared to visual quantifications of TF clusters. Third, expression amplification saturates at a rapamycin concentration (i.e., between 0.5 nM and 5.0 nM, Fig. 2b first two panels) that is lower than the saturation concentrations for the visual quantifications of TF clustering (e.g., ~100 nM for Fig. 1e). These data implicate that while TF clustering can amplify

gene expression, there is an upper bound for such amplification, and this upper bound is not trivially determined by either the size or the number of TF clusters (that we quantified). Altogether, these results support that TF clustering propensity can greatly influence target gene expression.

**Evidence for the direct activation and modulation of transcriptional dynamics by TF clustering**. We next sought for direct evidence supporting the activation of target genes by TF clusters. While we and others have previously used transiently transfected RNA stem-loop-based transcriptional reporters (e.g., PP7 or MS2) to visualize spatiotemporal gene activation by TF clusters[26,28], a direct support for the activation of stably integrated gene loci by TF clusters is lacking due to technical challenges. Importantly, our system offers a unique opportunity for addressing several critical questions regarding the gene regulatory role of TF clusters (Fig. 3a). First, because the formation of TF clusters does not depend on DNA binding and the DNA binding capability of the synthetic TF can be controlled by doxycycline (for rTetR), we could directly test whether rapamycin-induced TF clusters could bind to the genomic loci of integrated target reporter genes in a doxycycline-dependent manner. Second, because we could simultaneously visualize TF clusters and nascent transcription signals of target reporters in the same cell, we could explore their potential interactions spatiotemporally. Third, by tuning the TF clustering propensity with rapamycin, we could examine the gene regulatory role of TF clustering at the level of transcriptional bursting modulation by analyzing target genes' transcriptional dynamics at different rapamycin concentrations.

To investigate the locus-specific DNA binding ability of induced TF clusters, we carried out DNA fluorescence in situ hybridization (DNA FISH) assay to simultaneously image the reporter gene loci and the TF clusters. Briefly, Cy5-labeled DNA probes targeting the integrated reporter genes were generated by nick translation, which were hybridized to target reporter genes in the U2OS-7TetO cells, and the EGFP signal was preserved during the hybridization (see "Methods" for details). By performing three-color confocal imaging (EGFP for TF clusters, Cy5 for reporter gene loci, and DAPI for DNA staining) of the resulting cells at multiple z slices, we were able to detect co-localization events between TF clusters and reporter gene loci when doxycycline is present (i.e., when the DNA binding capacity of the synthetic TF is switched on) (Fig. 3b, Supplementary Fig. 2a–c). Importantly, the co-localization between the two signals depended on the DNA binding capability of the TF clusters, as almost no co-localization could be observed in the absence of doxycycline (Fig. 3c, Supplementary Fig. 2a). Furthermore, we found the intensity of the TF clusters co-localizing with reporter gene loci is significantly lower compared to the top three brightest non-co-localizing TF clusters (Supplementary Fig. 2d), indicating that the largest TF clusters might not possess the strongest DNA binding capability. Together, these results demonstrate that rapamycin-induced TF clusters can bind to DNA in a locus-specific and doxycycline-controllable manner, providing evidence for the causal gene regulatory role of synthetic TF clusters.

We next asked whether the locus-specific binding of TF clusters is functionally important, and if so, we would expect to observe spatiotemporal interactions between TF clusters and nascent transcription signals from stably integrated target gene loci, as the DNA locus-specific binding of TF clusters would result in the transcription of target genes. To test this hypothesis, we performed two-color time-lapse imaging of both nascent transcription signals from stably integrated target gene loci and signals from TF clusters in U2OS-7TetO cells, allowing us to

analyze their spatiotemporal interactions (Fig. 3a). Intriguingly, we found that TF clusters appear to display binding and unbinding dynamics at individual nascent transcription sites (Fig. 3d, e, Supplementary Fig. 3a, Supplementary Movie 1) with binding dwell times following an exponential-like distribution (Supplementary Fig. 3b), implicating the stochastic nature of TF cluster's unbinding from the reporter loci. Temporally, there are several potential modes of interactions between TF clusters and nascent transcription sites (Supplementary Fig. 3c). For example, for some reporter loci, we observed nascent transcription signals but not TF cluster signals, and for some other loci, we observed repeated entangling between both signals. It should be noted that these spatiotemporal analyses of interactions relied on the visual inspection of co-localization between two fluorescent signals from two different channels, which may not be accurate enough due to the diffraction-limited resolution of our microscope. Despite the potential limitation, these two-color dynamic results, together with the DNA FISH results, implicate that TF clusters can bind to reporter gene loci to activate the transcription of the target gene.

To further support that these spatiotemporal interactions result from the binding of TF clusters instead of random co-localization between two signals, we took snapshots of hundreds of cells in the presence or absence of doxycycline and analyzed the co-localization. In the presence of rapamycin and doxycycline (to enable DNA binding), we captured events where the two signals overlap at the same spatial localization inside the nucleus (Fig. 3f). To quantify such overlap more systematically, we first detected the pixel locations of all TF cluster signals in the EGFP channel, and then computed the corresponding mCherry (i.e., PCP) signal levels at the same locations (Supplementary Fig. 3d, e and "Methods"). By doing so, we found that there is approximately 4 percent of TF clusters co-localizing with nascent transcriptional signals (Fig. 3g). To estimate the by-chance co-localization percentage, we generated pseudo-clusters in the EGFP channel (Methods), and the same analysis above showed that there is smaller than 1 percent co-localization (Fig. 3g and Supplementary Fig. 3d). As an additional control, when the TF's DNA binding capability is switched off (i.e., without adding doxycycline), the same analysis yielded a low co-localization fraction for either real or pseudo TF clusters (i.e., smaller than 1 percent, Fig. 3g and Supplementary Fig. 3e). Conversely, when analyzing the fraction of PCP signals overlapping with TF signals, we also observed a significantly higher overlap rate compared to pseudo PCP signals (i.e., control, Supplementary Fig. 3f, g). An additional intriguing finding from these analyses is that among overlap signals there is a negative correlation between the intensities of TF clusters and nascent transcription sites (Supplementary Fig. 3h), corroborating with the result from DNA FISH assay (Supplementary Fig. 2d).

Of note, these results likely also suffer from the same limitation as noted above for the spatiotemporal analysis, i.e., light diffraction in our microscope precludes the analysis of spatial co-localization at a high enough resolution. As such, the co-localization between two signals indicates the spatial proximity between two biological entities, and may not suggest their actual interactions. Nevertheless, the spatial proximity between TF clusters and nascent transcription sites is statistically significant, consistent with the scenario that TF clusters can bind to reporter loci. Furthermore, the relatively low rate of overlap in the snapshot data could arise from at least two reasons. First, from the two-color temporal traces we found that a fraction of PCP transcription signals did not have visible TF clusters during their lifetime (Supplementary Fig. 3c), potentially because of TF-cluster-independent transcription activation (or we did not the catch the TF cluster). Second, even if PCP signals appeared to be activated by TF clusters, co-localization between PCP and TF signals could be detected for only about 10% the time during the

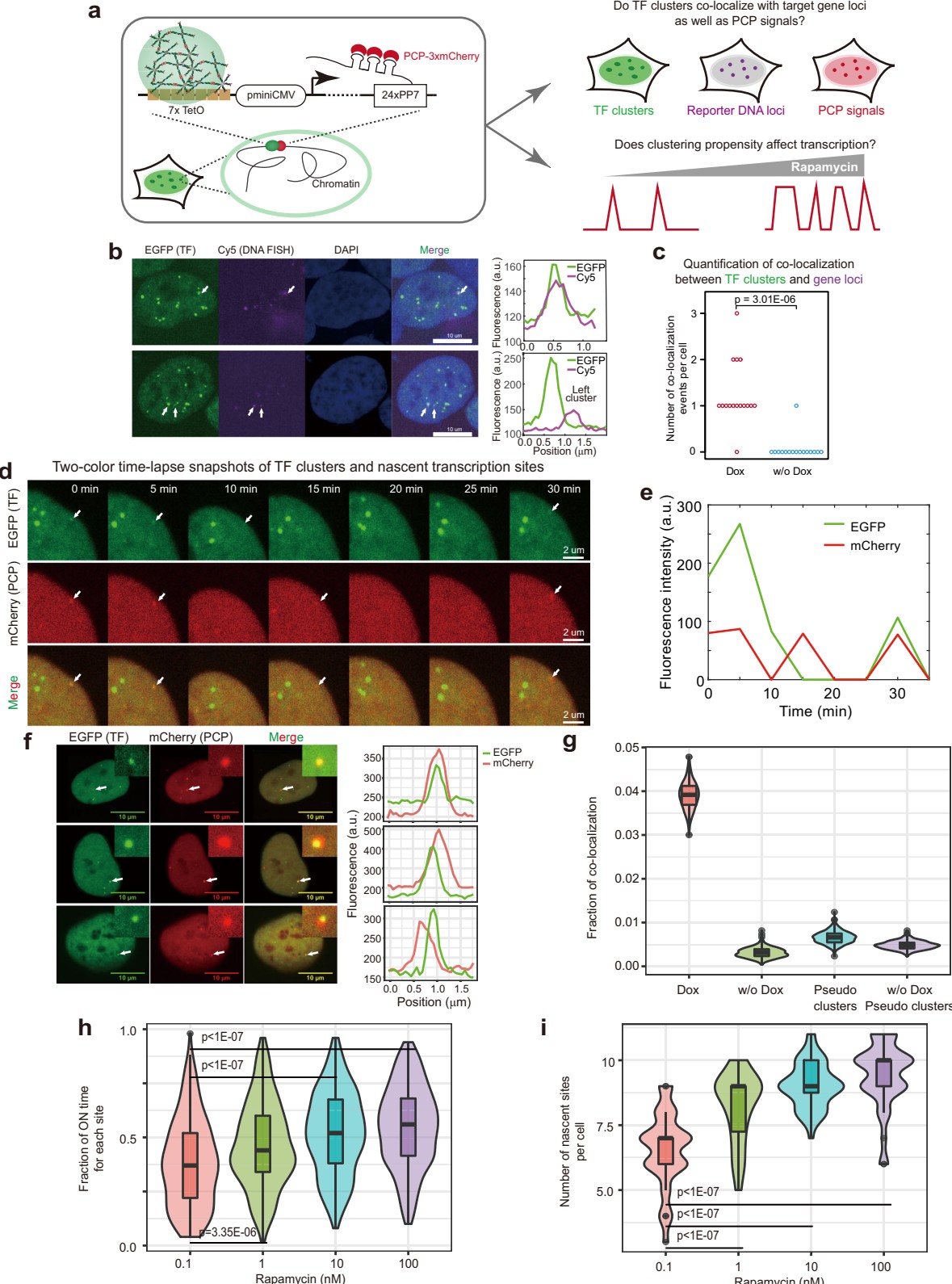

(b) EGFP (TF) | Cy5 (DNA FISH) | DAPI | Merge

(d) Two-color time-lapse snapshots of TF clusters and nascent transcription sites

(f) EGFP (TF) | mCherry (PCP) | Merge

(c) Quantification of co-localization between TF clusters and gene loci

(g)

(h)

(i)

lifetime of the transcription site (estimated based on the two-color traces).

Thus, together with the two-color time-lapse analysis, the significant overlap between the two signals in the presence of rapamycin and doxycycline in the snapshot data provides evidence consistent with the direct activation of target reporter transcription by TF clusters.

If TF clusters are indeed responsible for activating target transcription, we should observe significant changes in transcriptional dynamics when titrating rapamycin concentration (Fig. 3a). To test this hypothesis, we performed transcriptional imaging of the same cell line under four different rapamycin concentrations. We found that TF clustering propensity indeed significantly affects transcriptional dynamics (Supplementary Fig. 4a,

**Fig. 3 Evidence for the direct activation and modulation of target transcription by TF clustering. a** Assay design. Note that U2OS-7TetO cells were used for the following experiments. **b**–**c** DNA fluorescence in situ hybridization (DNA FISH) assay for analyzing gene locus-specific binding of TF clusters. Three-color images of cells with arrows indicating co-localization events between TF clusters and Cy5-labeled DNA probes targeting the reporter gene loci (**b**, left), and the corresponding fluorescence intensity line profiles (**b**, right). Dot plot showing the number of co-localization events per cell for two culture conditions, i.e., with or without 0.3 μg/mL doxycycline and both with 100 nM rapamycin (**c**, $n = 16$ cells for each condition and $p$ value from two-sided $t$ test). **d**–**e** Time-lapse analysis of spatiotemporal interactions between TF clusters and nascent transcription sites of reporter genes. Snapshots (**d**) and the intensity trajectories of arrow-indicated reporter locus (**e**). **f**–**g** Evidence supporting the direct transcriptional activation by TF clusters. Two-color images showing the spatial co-localization between TF clusters and nascent transcriptional sites and corresponding fluorescence intensity line profiles (**f**). The fraction of TF clusters co-localizing with nascent transcriptional signals were calculated for two culture conditions (with or without 0.3 μg/mL doxycycline, and both with 100 nM rapamycin) (**g**). Fraction of co-localization was calculated by bootstrap (resampled 1000 times with replacement), and more than 2500 TF clusters were used for each calculation. n = 615 (with dox) and 897 (without dox) cells. **h**, **i** Evidence supporting the direct modulation of transcriptional dynamics by TF clustering propensity. Cells were cultured at indicated rapamycin concentrations, and doxycycline (0.3 μg/mL) was added. Both the fraction of ON time for each nascent site (**h**) and the number of nascent sites per cell (**i**) vary with rapamycin concentration. Dot numbers are 236, 378, 362, and 348 from left to right in **h**. Cell numbers are 36, 46, 40, and 37 from left to right in **i**. $p$ values were calculated by one-way ANOVA and TukeyHSD. For boxplots, horizontal line indicates median and box ranges from first to third quartile, with whiskers extending up to 1.5 × interquartile range. Source data are provided.

Supplementary Movie 2). At the individual gene locus level, increasing clustering propensity greatly increases the fraction of time that the nascent transcription site is detected (Fig. 3h), and reduces the time to the first transcriptional burst post-doxycycline addition (Supplementary Fig. 4b). Because the cell line contains ~11 integrated copies of the same reporter gene (estimated by the maximum detected nascent sites and the DNA FISH result), we found that increasing TF clustering propensity leads to a more coordinated activation of genes of the same regulon, as more actively transcribing gene loci were observed inside a cell (Fig. 3i).

Together, these data provide imaging-based evidence indicating the direct and enhanced activation of target gene transcription by TF clusters. The data also suggest that cells could activate more genes in a specific regulon by enhancing the clustering propensity of the TF without altering its expression level.

**High TF clustering propensity enables bimodal target responses.** Because clustering-prone TFs often play key roles in cell fate determination, we next asked whether the clustering of TF could confer emergent behaviors that are often observed in canonical cell fate control systems[62].

A typical emergent behavior in a system is the presence of bimodal state distributions[63]. We thus characterized the distributions of both input and output signals in our system, namely TF fluorescence signals and reporter protein fluorescence signals. By using flow cytometry to characterize U2OS-7TetO cells, we found that the input signals are unimodally distributed across varying rapamycin concentrations (Supplementary Fig. 5a). In contrast, the output signals are bimodally distributed at higher rapamycin concentrations (Supplementary Fig. 5b). We used bimodality index[64] to quantify these distributions, and found that the bimodality index for the output signals displays a non-monotonic dependence on rapamycin concentration, whereas the bimodality index for the input signals is constantly low (Supplementary Fig. 5c). These data indicate that the gene regulation function could be modulated to enable bimodality when TF clustering propensity is high.

We speculated that the modulation of both the effective TF binding affinity and the ultrasensitivity of the gene regulation function could be responsible for the emergence of bimodality (Fig. 4a). To test this, we fitted the input–output data from flow cytometry with a Hill function (Methods) across a range of rapamycin concentrations (Fig. 4b, Supplementary Fig. 5d). We found that as rapamycin concentration increases, the effective binding affinity of the TF to DNA generally increases (i.e., dissociation constant decreases) (Fig. 4c, Supplementary Fig. 5e) as well as the Hill coefficient (Supplementary Fig. 5f).

Because we only captured part of the dose–response curve for the low rapamycin conditions, we needed to rule out the potential artifact on fitted Hill coefficients due to such sampling bias. First, we carried out the fitting at a range of fixed Hill coefficients, and found that the squared norm of the residual reached minimal at varying Hill coefficients under different rapamycin concentrations (Supplementary Fig. 5g), as expected. Second, we quantified the local sensitivity of the response curve by linear fitting (in log-log scale) and reassuringly found that the local sensitivity (i.e., slope) increases as rapamycin concentration increases (Supplementary Fig. 5h). Together, these results demonstrate that both the effective binding affinity and the ultrasensitivity of the gene regulation function are modulated by TF clustering propensity, and the effective binding affinity appears to be modulated by a much larger extent compared to the ultrasensitivity, implicating that TF clustering has a stronger influence on the sensitivity than the ultrasensitivity of the gene regulation function in our system.

The data prompted the question regarding how the modulation of the two parameters (of the gene regulation function) each contributes to bimodality in the above system. It is apparent that in addition to the range of input signals, the decreasing $K_d$ value as rapamycin increases is important for the appearance of bimodality. Yet, the contribution from the increase in Hill coefficient is less obvious. To examine this, we focused on conditions where $K_d$s are relatively stable while Hill coefficients are changing, i.e., between 5 nM and 30 nM rapamycin (Supplementary Figs. 5e, f boxed region). We observed that among these conditions, the bimodality index increases when Hill coefficient increases (Supplementary Fig. 5c), indicating that the modulation of ultrasensitivity likely contributes to the appearance of bimodality, together with the modulation of the effective binding affinity.

Furthermore, these modulations effectively lead to an increase in the absolute cell-to-cell variability (i.e., standard deviation) in the output signal, whereas the variability of the input signal remains relatively constant across all conditions (Supplementary Fig. 5i). In contrast, the relative cell-to-cell variability (i.e., coefficient of variation or noise) displays a non-monotonic behavior (Supplementary Fig. 5j), suggesting that TF clustering can modulate gene expression noise. It should be noted that the way how TF clustering modulates absolute or relative cell-to-cell variability depends on the range of TF expression levels in the cell population.

Similar modulations of gene regulation function by TF clustering were also observed in a different host cell line (i.e., CHO) (Fig. 4d, Supplementary Fig. 6a–c), whose output distributions can also become bimodal (Supplementary Fig. 6d). Note that local sensitivity analysis on the CHO-Gal4 data also supported the

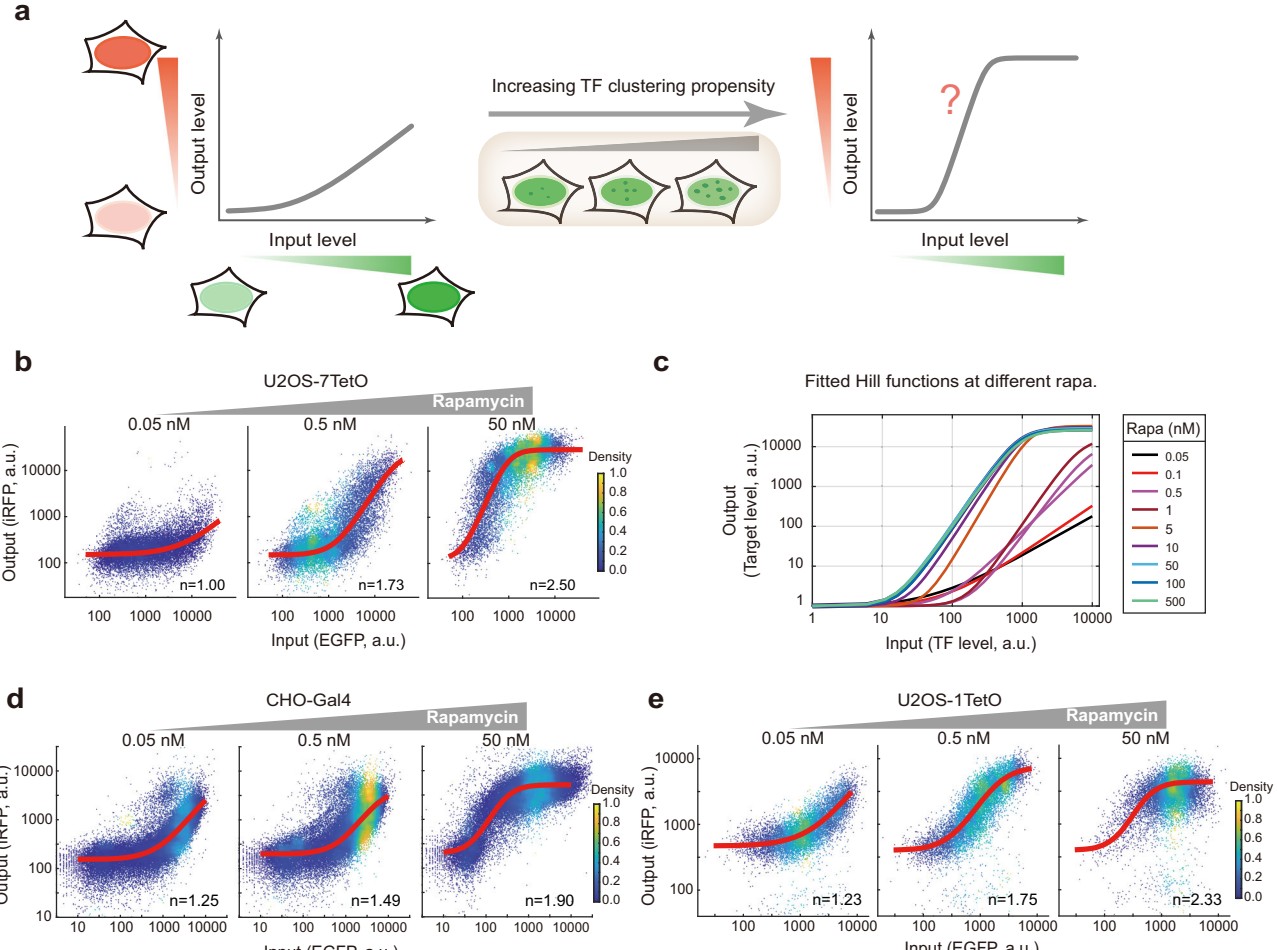

**Fig. 4 TF clustering propensity modulates the gene regulation functions of synthetic TFs. a** Assay design. **b-c**, TF clustering propensity modulates the gene regulation function of the U2OS-7TetO system. Steady-state TF (EGFP) and reporter (iRFP) signals (i.e., inputs and outputs) were quantified by flow cytometry at indicated rapamycin concentrations (with 0.3 μg/mL doxycycline) and were fitted by Hill function (red curve) (**b**). *n* indicates fitted Hill coefficient. See also Supplementary Fig. 5d–f for more details. Fitted Hill functions were plotted together to indicate the modulation of both effective binding affinity and Hill coefficient (**c**). **d-e** TF clustering propensity also modulates the gene regulation functions in CHO-Gal4 (**d**) and U2OS-1TetO (**e**) systems. Analogous experiments and fitting as in (**b**) were performed for these two systems (see Methods). See also Supplementary Fig. 6. Source data are provided.

modulation of sensitivity by TF clustering (Supplementary Fig. 6e). Thus, these data demonstrate that TF clustering-mediated tuning of the gene regulation function can lead to switch-like responses in the target gene, bimodal output distributions, as well as modulated cellular heterogeneity.

Because the reporter contains seven TF binding sites in the U2OS-7TetO cell line, we next addressed whether multiple binding sites (in addition to TF clustering) are necessary for conferring ultrasensitivity. In other words, it is possible that ultrasensitivity might not occur without tandem TetO repeats even when TF clustering propensity is high. We thus analyzed the input–output functions in the U2OS-1TetO cell line, whose reporter gene promoter contains only one TF binding site. We found that the ultrasensitivity and effective binding affinity are similarly modulated by rapamycin as in the U2OS-7TetO cell line, and the Hill coefficient increases to up to ~2.4 (which is relatively smaller than the preceding cell line) (Fig. 4e, Supplementary Fig. 6f–i). This result suggests that TF clustering can contribute to ultrasensitive gene regulation function independent of tandem binding sites, and the presence of tandem binding sites could further enhance the ultrasensitivity.

**Sustained transcriptional response to transient stimuli conferred by TF clustering.** Given the large degree of modulation of the gene regulation function by TF clustering, we asked whether the system could exhibit additional emergent behaviors that are common in canonical cell fate control systems such as memory, i.e., sustained response to transient input (Fig. 5a).

To test the presence of sustained response, we sought to synthesize a transient expression pulse of the TF, mimicking the pulse of regulator concentration or activity during cell fate transition in some natural systems. To do so, we used the preceding CHO cell line whose TF expression is under inducible doxycycline control. We then generated a pulse of TF expression by transient administration of doxycycline (i.e., at 0.1 μg/mL for 12 h) (Fig. 5b). To probe the effect of TF clustering, we compared two culture conditions, i.e., without or with rapamycin (50 nM). Both TF and reporter fluorescence signals were quantified by flow cytometry throughout the time course of the experiment (Fig. 5c, d). Note that we gated cells based on the input signals in order to ensure comparable input signal distributions between conditions at different time points (Methods and see Supplementary Fig. 7a for ungated data).

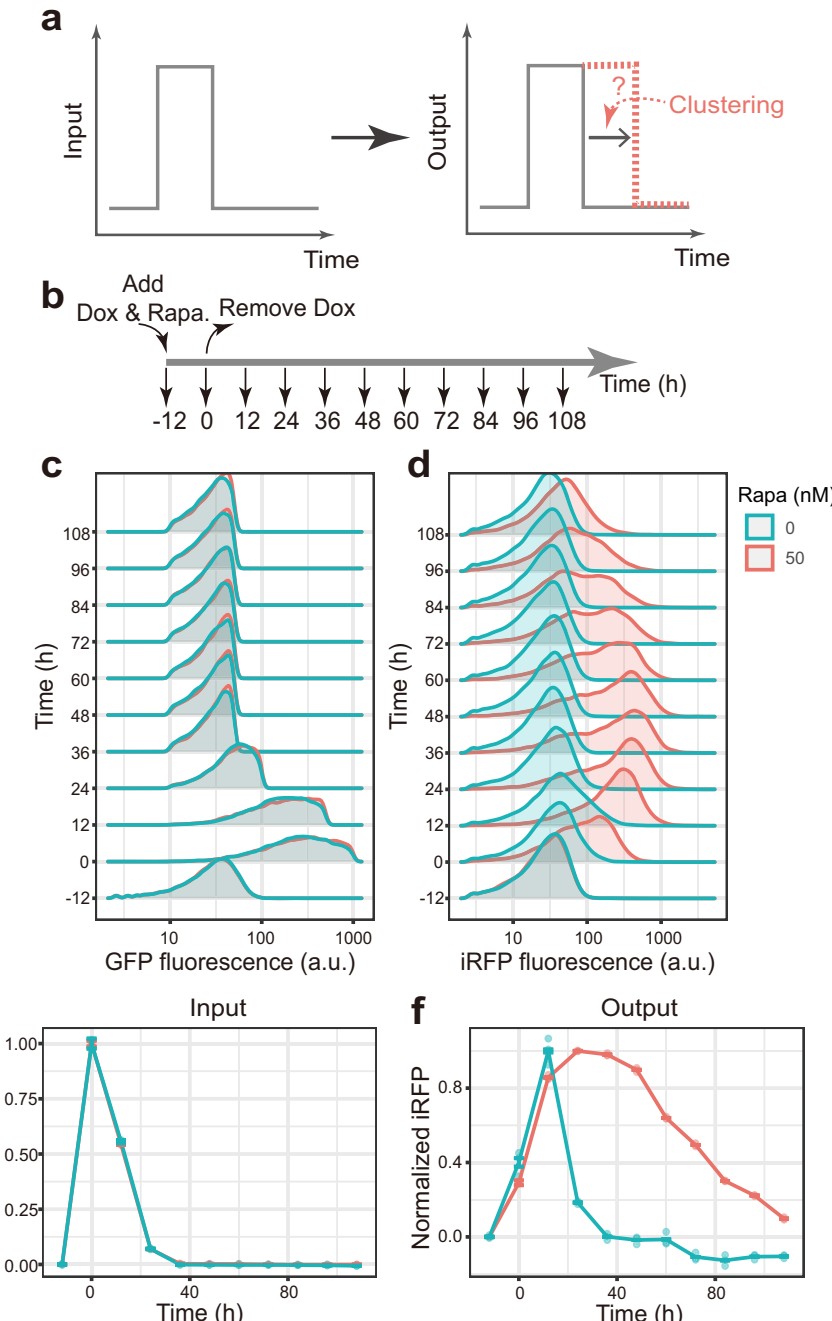

**Fig. 5 TF clustering confers sustained transcriptional response to transient stimuli. a** Assay design. **b-d**, TF clustering results in a sustained temporal response of the target gene. CHO-Gal4 cells were subjected to a transient doxycycline (0.1 μg/mL) treatment, which induced the expression of the synthetic TF, Gal4-VP64. Both the TF and the reporter fluorescence levels were then quantified by flow cytometry at indicated time points (**b**). Rapamycin (50 nM) was present or absent throughout the time course. The TF levels display a transient increase for both conditions (**c**), whereas the reporter levels exhibit distinct behaviors between the two conditions (**d**). Note that we gated the TF intensity to ensure comparable input levels. See Supplementary Fig. 7a for ungated results. Data from three biological replicates were pooled together for (**c**, **d**), with each replicate behaving similarly. **e-f** Normalized input (**e**) and output (**f**) traces showing that while the input signals between the two conditions are almost identical, the output signal displays a sustained transcriptional response when TF clustering propensity is high (50 nM rapamycin). Data are presented as mean ± S.D. (n = 3 biological replicates). See Supplementary Fig. 7b, c for ungated results. Source data are provided.

To detect potential sustained response behavior, we compared the temporal input–output relationships between the two different rapamycin conditions. By plotting the normalized input trajectories (i.e., normalized mean TF signals over time) and the normalized output trajectories (i.e., normalized mean reporter signals), we found that an input pulse was successfully synthesized for both conditions, and the two input pulses share

similar temporal profiles (Fig. 5e, Supplementary Fig. 7b). In contrast, the output trajectories are drastically different between the two conditions (Fig. 5f, Supplementary Fig. 7c). More specifically, when the TF clustering propensity is high (i.e., at 50 nM rapamycin), the reporter signal decays much slower compared to the condition without rapamycin (Fig. 5f). To test whether the sustained response indeed occurs at the level of

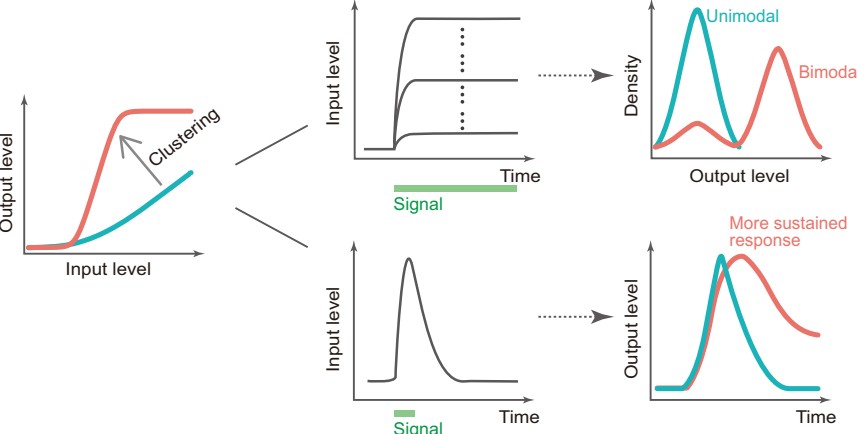

**Fig. 6 A summary of the modulation of the gene regulation function by TF clustering and the resulting behaviors of the system.** TF clustering modulates the gene regulation function by tuning both the effective binding affinity and the ultrasensitivity (i.e., Hill coefficient) (left), leading to bimodal target response (output) at steady state (top right). When subjected to a transient signal, TF clustering confers a sustained transcriptional response of the target gene, whereby the output signal decays much slower compared to the scenario without TF clustering (bottom right). Note that the effective binding affinity is modulated by a larger extent compared to the ultrasensitivity.

transcription (and not at the level of protein translation or stability), we quantified the mRNA levels of iRFP during the time course and reassuringly found that at high clustering propensity the mRNA level decays much slower compared to the no rapamycin condition (Supplementary Fig. 7d).

In addition to flow cytometry quantification, we performed analogous experiments using time-lapse imaging, and we similarly observed a slower decay of reporter signals at higher rapamycin concentrations (Supplementary Fig. 7e–g). Note that the number of visible TF clusters decreased over the time course (Supplementary Fig. 7h). Thus, in response to pulsed-like input signals of similar time scales, the system appears to "memorize" the input signal for a longer time when the TF clustering propensity is high. In line with the quantifications at the gene expression level, we found that the promoter region of the reporter genes was acetylated to a higher level when the TF clustering propensity is high (Supplementary Fig. 8).

We next explored the potential mechanism underlying such sustained response behaviors. To do so, we first examined the distributions of input and output signals using the flow cytometry data along the experimental time course (Fig. 5c, d). We found that with similar input distributions, the output signal distributions differ greatly for the two rapamycin conditions. More specifically, when TF clustering propensity is high, the output signal becomes bimodally distributed during the time course, consistent with the preceding observation of bimodal distributions when the gene regulation function is modulated. In contrast, the output signal for no rapamycin condition is unimodally distributed. Note that the shape of the output distributions should depend on the range of the input levels.

These data suggest a sequence of events constituting the sustained response behavior when rapamycin is present. More specifically, because of the increased effective binding affinity at high TF clustering propensity (Supplementary Fig. 9a), transient upregulation of TF expression allows the reporter genes to switch on when the TF concentration crosses a much lower threshold (i.e., dissociation constant) compared to the condition without rapamycin (Supplementary Fig. 9b). Due to the cell-to-cell variability in TF expression and the modulated gene regulation function, output signal displays bimodal distribution when rapamycin is present. As the pulse decays, the TF concentration then drops below the same threshold level and the reporter genes switch back to the off state. Therefore, because the threshold is

lower in the presence of rapamycin, the output signal lasts for a longer duration compared to the condition without rapamycin (Supplementary Fig. 9b). Of note, in the above scenario we described gene control in a binary fashion for simplicity.

To validate this picture, we first carried out computational simulations to illustrate the role of modulated gene regulation function, which successfully recapitulated the sustained response behavior (Supplementary Fig. 9c, d). To determine which parameter modulation (i.e., effective binding affinity or Hill coefficient) is important for the observed behavior, we performed two additional simulations by artificially fixing either the effective binding affinity or the Hill coefficient, in which we assumed that one of the two parameters of the gene regulation function measured in the condition with high TF clustering propensity remained the same as the no rapamycin condition. By doing so, we found that the modulation of the effective binding affinity, but not the modulation of the ultrasensitivity, contributes to the sustained response behavior (Supplementary Fig. 9e, f). These results indicate that the effective binding affinity of the gene regulation function was greatly modulated by TF clustering, leading to the observed sustained response (i.e., memory-like) behavior.

## Discussion

A growing list of studies have indicated the potential cell fate-determining roles played by TF clusters[7–13,15,16], yet the quantitative roles of TF clustering in gene regulation remained poorly understood. In this work, by rationally designing a bottom-up synthetic gene regulatory system with chemically tunable TF clustering, we delineated the roles of TF clustering in the activation and amplification of gene transcription, and the modulation of gene regulation function. The observed "emergent" behaviors at high TF cluster propensity, including bimodality and sustained response behavior (Fig. 6), resemble phenomena observed in classical cell fate control systems[23,63,65]. This is consistent with the picture that TF clustering could play a key role in some cell fate control systems[20,21].

Our synthetic system enables unique opportunities for establishing the causal relationship between TF clustering propensity and the gene regulatory role of clustered TFs at multiple levels. At the level of target gene output, we titrated the clustering propensity of the TF with rapamycin, and at the meantime, measured the associated changes in target gene's responses. Intriguingly, apart from observing transcriptional amplification by TF

clustering, which is in line with several recent studies[26–28], we found that such amplification might occur prior to the appearance of visually detectable TF clusters, and that the degree of amplification is not trivially determined by the size or number of visible clusters. At the level of DNA binding, we showed that TF clusters exhibit spatial co-localization with reporter gene loci in a doxycycline-dependent manner, implicating the locus-specific binding ability of synthetic TF clusters. At the level of transcriptional activation, we presented spatiotemporal analyses of interactions between TF clusters and nascent transcription sites to illustrate the interactions between the two signals, consistent with the model of the direct activation of reporter transcription by TF clusters. This finding was substantiated by further evidence showing the modulation of transcriptional bursting dynamics by TF clustering propensity. Notably, while there appeared to be a negative correlation between TF cluster intensity and nascent transcription site intensity (Supplementary Fig. 3h), questions remain as for how the size of the TF cluster is quantitatively linked to target transcriptional activity[66]. Of note, our co-localization analysis in both the snapshot data and the temporal data could suffer from limitations arising from limited spatial resolution and may thus fail to distinguish between co-localization events with or without physical interactions.

Through systematic dissections, we demonstrated that parameters of the gene regulation function can be quantitatively tuned by TF clustering. While the tuning of ultrasensitivity can be achieved by TF oligomerization[56,67], a concurrent tuning of both effective TF binding affinity and ultrasensitivity of the gene regulation function indicates the unique capabilities of rapamycin-mediated TF clustering. It is intriguing to note that TF clustering confers a stronger modulation on the effective TF binding affinity than on the Hill coefficient for all three systems characterized (when measured by the relative fold change of the fitted parameter). Such a differential modulation of the parameters may arise from the largely enhanced sequence-specific DNA binding capabilities of the TF clusters, which is consistent with the findings that rapamycin significantly speeds up the appearances of target genes' initial transcription events and increases the number of activated gene loci. The relatively weaker but consistent modulations of ultrasensitivity across three synthetic TFs are in line with the picture that TF clustering can confer nonlinear behaviors[28,68]. Yet, the observed differential modulation raises open questions regarding the functional principles of TF clustering: whether TF clustering affects gene regulation mainly through the change in the effective TF binding affinity (or sensitivity) or if there are additional modes of regulation (e.g., by mainly modulating ultrasensitivity).

We demonstrated that the modulation of the gene regulation function can lead to emergent behaviors in the system at high TF clustering propensity, including bimodality and sustained response behavior. While the emergence of bimodality appears to be a trivial consequence resulting from the modulation of the effective TF binding affinity, the fact that TF clustering can significantly tune the effective binding affinity is striking, which could provide insights into the many bimodally distributed genes in the transcriptome[64]. From the data, we also observed that the tuning of ultrasensitivity could synergize with the tuning of effective binding affinity to confer bimodality. Yet, the relative contributions remain to be quantitatively determined. Another emergent behavior when TFs are prone to cluster is sustained response (i.e., memory-like) behavior, in which the increase in effective TF binding affinity allows a relatively sustained activation of the target reporter gene in response to a short input pulse. Yet for this memory-like behavior, we cannot rule out potential contributions from other parts of the system, including the contributions from the chromatin environment.

It is unclear whether the conclusions from synthetic TFs could be generalizable to natural TFs. A direct test of the role of TF clustering in natural systems would require tuning the clustering propensity of natural TFs analogously to our synthetic system. Despite the lack of direct evidence, it is still intriguing to speculate what advantages a cell fate control system based on natural TF clustering would offer compared to a canonical control system composed of a circuit of interacting components. It is possible that the clustering of a master cell fate regulator can synergize with circuit-based feedback interactions to enhance the performance of the cell fate control system.

This work highlights the advantages of synthetic biology approaches for dissecting complex gene regulatory mechanisms[56]. The design principles learned from our synthetic system could help to elucidate the emergent functions conferred by protein clustering in diverse biological contexts[1,2], including disease, development, and immunity. More importantly, our results reveal a potentially wide-spread application of small molecule-based tunable clustering mechanism, which could be implemented for controlling cell states or for interfering with native control systems.

## Methods

**Plasmid construction**. Plasmids used for mammalian cell transfection were based on the PiggyBac transposon system. For constructing synthetic TF or reporter plasmids, backbone vectors were linearized by restriction endonucleases (NEB, see also Supplementary Table 1 for details of reagents), and the insert DNA fragments were either PCR amplified from genomic/plasmid DNA or assembled from synthesized oligos using PrimeSTAR Max DNA Polymerase (TAKARA, R045B). All plasmids were constructed by Gibson assembly and verified by Sanger sequencing (RUIBO). Plasmids were replicated in DH5α (CWBIO, CW0808S) cells using standard protocols. Plasmid maps are available upon request from the corresponding author.

**Cell culture**. U2OS (ATCC) cells were cultivated in Dulbecco's Modified Eagle Medium (Gibco), supplemented with 10% fetal bovine serum (FBS, Gibco) and 1% penicillin-streptomycin (Sigma-Aldrich). The culture medium for Chinese Hamster Ovary (CHO) cells (obtained from the ATCC) includes RPMI 1640 media (Gibco), 10% FBS (Gibco), and 1% Pen-Strep. All cell cultures were kept under 5% $CO_2$ and 37 °C temperature. The cell culture was changed daily and cells were passaged every three days.

**Cell transfection**. All plasmids were transfected into cells by liposome-based transfection using Lipofectamine®LTX & PLUS™ (Invitrogen), except the plasmid encoding PCP-3xmCherry (for imaging nascent transcription sites), which was transduced by lentivirus. For liposome transfection, the relative ratio between plasmid (µg), PLUS™ (µL), and Lipofectamine®LTX (µL) was set at 1:1:3. More specifically, cells were plated into wells in a 24-well plastic-bottom plate 12–24 h before transfection. At the time of transfection, the culture typically researched a cell line-specific confluency level, i.e., ~60% for CHO cells and ~90% for U2OS cells. For a typical 24-well transfection, 0.8 µg plasmids together with 0.8 µL PLUS reagent and 2.4 µL LTX were mixed and added to the cell culture. After 6–8 h of U2OS transfection, and 18-24 h after CHO transfection, the cell culture medium was replaced with fresh medium.

**Construction of stable cell lines**. The construction of monoclonal or polyclonal cell lines was based on the preceding transfection protocol followed by antibiotic selection and flow sorting. More specifically, PiggyBac transposon was used for the integration of foreign plasmids into U2OS or CHO cells. A plasmid encoding piggyBac transposase was co-transfected with plasmids carrying synthetic TF or reporter gene to enable chromosomal integration. The mass ratio of the piggyBac plasmid is 1/5 of the total plasmid mass. 48 h post-transfection by LTX, cells were cultured in a medium containing antibiotic (100 µg/mL Hygromycin B or 5 µg/mL Puromycin Dihydrochloride) in order to select for cells with successful integrations of the plasmid of interest. After antibiotic selection for 1-2 weeks, doxycycline (0.1 µg/mL, 24 h before sorting) was added to induce the reporter gene, and cells carrying the positive reporter signals were deposited into 96-well plates using fluorescence-activated cell sorting (FACS). The deposited single cells were then cultured and expanded in doxycycline-free medium. The culture medium was changed every 5 days after sorting. Monoclonal cell lines were screened and selected using the fluorescence microscope. For screening, we looked for cells with multiple integration of reporter genes (identified by nascent transcription sites) or for cells with an appropriate synthetic TF concentration. For polyclonal cell lines, a

similar procedure was used except that a population of cells (instead of single cells) were deposited into the same well during flow sorting.

**Characterization of chemically induced TF clusters**. Prior to imaging, monoclonal U2OS-7TetO cells were plated in a 24-well glass-bottom culture plate (24-well laser confocal culture plate, Cellvis), and the plating density was controlled such that the culture would reach ~60% confluency at the time of imaging. 12 h after the inoculation of U2OS cells, 0.3 μg/mL doxycycline (Clontech) and varying concentrations of rapamycin (Harvey), i.e., 0 nM, 0.05 nM, 0.1 nM, and all the way up to 1000 nM, were added into the culture medium. After another 12 h, images of transcription factors (EGFP, excitation laser: 488 nm, emission filter: 502–540 nm) were captured (9 z slices to cover the entire cell) on a spinning disk confocal microscope (Andor Dragonfly) using a Plan Apo Lambda 100x/1.44 oil objective (Leica). Fluoro Brite DMEM medium (Thermo) were used during optical imaging experiments. Live-cell imaging was conducted under 5% $CO_2$ and 37 °C humidified air.

For the analysis of TF clusters, max intensities of each pixel across z slices were calculated by ImageJ. Custom Matlab (MathWorks) codes were used to automatically segment individual cells and to identify TF clusters within each cell. For the identification of TF clusters, a log filter of the maximum intensity-projected image was first performed, and local maxima were identified. The size of the TF cluster was estimated based on the properties of the pixels surrounding each local maximum. More specifically, the size of the TF cluster was defined by a circle (the diameter of which was reported as the cluster size) around the local maximum, such that the mean pixel intensity inside the circle is between 1.2× to 1.8× of the mean cellular fluorescence intensity. The identified TF clusters were manually inspected and used for downstream analysis. See Supplementary Fig. 1a for example outputs of the algorithm. It should be noted that the heterogeneous clustering behavior of cells under the same rapamycin conditions was unlikely due to inaccuracies of the algorithm.

**DNA fluorescence in situ hybridization (DNA FISH) assay**. To prepare template DNA fragment for probe generation, TRE3G-iRFP-24xPP7 plasmid (the plasmid used for generating integrated reporters) was linearized by restriction enzymes SpeI and NotI. A 3423 bp DNA fragment was purified by gel purification. This template DNA fragment was then used for generating labeled DNA probes by nick translation, with protocols adopted from literatures[69,70]. Briefly, in a 1.5 mL microcentrifuge tube on ice, we mixed 5 μL nick-translation reaction buffer (10×), 5 μL dNTP solution containing dTTP, dCTP, and dGTP, each at a concentration of 0.5 mM, 2.5 μL of 1 mM Cy5-dUTP, 2 μL of 15.6 U/mL DNase I, 2 μL of 10,000 U/ml E. coli DNA Pol I, with 1 μg template DNA fragment, and added nuclease-free water to a total volume of 50 μL. The reaction was incubated for 2 h at 15 °C. After incubation, 2 μL of 0.5 M EDTA was added to stop the reaction. The labeled DNA probes were purified by the HiPure Gel Pure DNA Micro Kit (Magen D2110). To the purified probes, 100 μg of salmon sperm DNA for 10 μg of DNA probes was added, which was then stored in −20 °C. Before usage, probes were denatured by incubating 0.3 μg probes in 15 μL hybridization buffer (for use on one coverslip of cells) for 5 min at 95 °C. Probes were then pre-annealed at 37 °C for 30 min before overnight hybridization.

For probe hybridization and imaging, U2OS-7TetO cells were seeded in 6-well plates with cover glass placed inside the well. To compare conditions with and without doxycycline, 0.3 μg/mL doxycycline and 100 nM rapamycin or 100 nM rapamycin only was added to the well. After culturing for 24 h, cells were washed with DPBS, and fixed with 2% paraformaldehyde in 1× PBS (pH 7–7.4) for 10 min at room temperature (RT). After three rinses with 1x PBS, cells were permeabilized with iced-cold 0.4% Triton-X-100 in 1× PBS for 5 min on ice. After another three rinses with 1× PBS, cells were incubated with 10 U/μL RNase I in 1× PBS for 1 h at 37 °C. After three rinses with 1× PBS, cells were further permeabilized with iced-cold 0.7% Triton-X-100 in 0.1 M HCl for 10 min on ice. After three rinses with 1x PBS, cells were denatured in 1.9 M HCl for 30 min at RT. After three rinses with ice-cold 1x PBS, cells were hybridized with probes overnight at 37 °C in a dark and humid chamber. For hybridization, each coverslip was placed with cell-side down onto 15 μL probe solution on a glass slide, sealed with rubber cement. After overnight incubation, rubber cement was carefully removed and coverslips were first rinsed with 2x SSC, placed in wells containing 2× SSC, and then washed for 30 min at 37 °C in dark with shaking. A second wash was carried out in 2× SSC for 30 min at RT in the dark with shaking, followed by a third wash in 1× SSC for 30 min at RT in the dark with shaking. Cells were then stained with DAPI solution for 5 min, and were mounted in one drop of Prolong Diamond Antifade Mountant (Thermo Fisher) on a slide. Z-stack imaging was performed with a spinning disk confocal microscope (Andor Dragonfly).

**Co-localization analysis between TF clusters and nascent transcription sites**. Monoclonal U2OS-7TetO cells were plated on a 24-well glass-bottom culture plate for 12 h. Doxycycline (0 or 0.3 μg/mL) and rapamycin (100 nM) were added into the cell culture to induce TF clustering and target gene expression. After another 12 h, confocal (Andor Dragonfly) images (with a ×100 objective) of TFs (EGFP, excitation laser: 488 nm, emission filter: 502–540 nm) and reporter genes' nascent transcriptional sites (bound by PCP-3xmCherry, excitation laser: 561 nm, emission

filter: 572.5–615.5 nm) were acquired using similar conditions as the preceding section. Acquired images were maximum intensity projected and processed as described above to detect TF clusters. For time-lapse imaging, the frame rate was 1 frame per 5 min.

To analyze snapshot data, for each TF cluster site, mCherry signals at the matched location were calculated and normalized with the cellular mean mCherry intensity. Normalized mCherry signals that were above a predefined threshold (i.e., 1.15) were considered as nascent transcription sites. This threshold was determined based on the distribution of normalized mCherry signals, and was justified by manual inspections of the above-threshold signals. A total of 2500 TF clusters were chosen from each condition for co-localization analysis.

To control for by-chance co-localization in the dox-containing condition, we randomly selected five EGFP sites inside each cell, and defined them as pseudo TF clusters. The matched mCherry signals were quantified as above.

To calculate the fraction of co-localization, the data were resampled 1000 times with replacement, and the bias-corrected 95% confidence intervals of co-localization fraction were calculated in each data set.

**Analysis for the modulation of transcriptional dynamics by TF clustering**. Monoclonal U2OS-7TetO cells were plated on a 24-well glass-bottom culture plate for 12 h. Then a gradient of rapamycin concentrations (i.e., 0.1 nM, 1 nM, 10 nM, 100 nM) was added to the culture media to induce TF clustering. After another 12 h, doxycycline (0.3 μg/mL) was added into cell culture and time-lapse images were collected at the same time. Time-lapse microscopy was performed on an automated inverted microscope (Nikon Ti-E) under 5% CO2 and 37°C humidified air using a Nikon Plan Apo Lambda ×40 objective (with a 1.5x magnifier). mCherry fluorescence (White LED light source; excitation filter: 560/40 nm, emission filter: 630/75 nm) images were taken every 10 min at five z slices for a total of 500 min time course. To analyze transcriptional dynamics, max intensities of each pixel across z slices were first calculated by ImageJ. A Matlab program with graphical user interface was used for semi-automatic identification and tracking of each nascent transcription site throughout the time series. It is noted that the number of nascent sites varies between single cells and across rapamycin concentrations.

**Flow cytometry quantifications**. Flow cytometry was used to capture the expression levels of both TFs and downstream reporter genes in single cells. These data allowed us to quantify the modulation of transcriptional activation by TF clustering. More specifically, cells were seeded on a 12-well plastic-bottom cell culture plate. The plating density was controlled such that the culture would reach ~90% confluency at the time of flow cytometry analysis. 12 h post-plating, doxycycline and varying concentrations of rapamycin were added into the culture medium. For U2OS cells, 0.3 μg/mL doxycycline was used unless specified. For input–output function characterizations of CHO-Gal4 system, a gradient of 9 doxycycline concentrations (1 to 10 ng/mL, except 9 ng/mL) in order to generate a large range of synthetic TF expression levels. To reach steady state, cells were cultured for 72 h, and the culture medium and chemicals were replaced every day. Fluorescence signals were collected on a BD Fortessa SORP flow cytometer.

**Fitting of the input–output function**. To facilitate robust fitting of the input–output data, flow cytometry data were binned according to the input (EGFP) levels. Cells within one log(input) interval were binned. More specifically, for input level within 100 a.u., the bin width is 10 a.u., and for the input level between 100 and 1000, the bin width is 100 a.u., etc. After binning, the input and output signals for cells within each bin were averaged and were then used for fitting with the following Hill equation using Matlab:

$$\text{Output} = \alpha_0 + \frac{\beta[TF]^n}{K_d{}^n + [TF]^n} \tag{1}$$

Analogous data binning process was implemented for the linear fitting of local sensitivity in the log-log scale.

**Characterization of sustained response by flow cytometry**. To generate a transient pulse of synthetic TF concentration, we used monoclonal CHO-Gal4 cells, whose synthetic TF expression is under doxycycline inducible control. Cells were first plated in a 12-well plastic-bottom plate. Doxycycline (0.1 μg/mL) and rapamycin (0 or 50 nM) were added to the cell culture. 12 h post-induction, the medium was replaced and doxycycline was removed. Cells were collected at indicated time points (i.e., Fig. 5b) for flow cytometry analysis. Gating on the EGFP channel was performed to ensure comparable TF levels between two rapamycin conditions. Gates were chosen based on the level of EGFP signal at each time point (1–100 at −12 h, 10–1000 at 0 h, 10–500 at 12 h, 10–100 at 24 h, and 10–50 at other time points). Ungated data was also shown.

**Characterization of sustained response by time-lapse imaging**. CHO-Gal4 cells were first plated on a 24-well glass-bottom culture plate for 12 h. Doxycycline (0.1 μg/mL) and rapamycin (0, 5 nM or 50 nM) were then added to the cell culture. 12 h post-induction, the medium was replaced and doxycycline was removed. Immediately following medium switching, cells were imaged on an automated

inverted microscope (Nikon Ti-E) under 5% $CO_2$ and 37 °C humidified air using a Nikon Plan Apo Lambda 40×objective. EGFP (excitation filter: 470/40 nm, emission filter: 525/50 nm) and iRFP (excitation filter: 650/45 nm, emission filter: 720/40 nm) fluorescence images were taken every hour for a total of 150 h. During imaging, medium was replaced every 24 h. A total of 19–20 field-of-views (i.e., replicates) were chosen for each culture condition. To exact the temporal input–output curves, an input or output signal was calculated for each field of view across time points. More specifically, for each image, pixels within the lowest 25% intensity values were regarded as background pixels. The mean intensity value of the highest 25% pixels, which were regarded as the pixels with signals, was subtracted by the mean background pixel value, and the resulting value was reported as the signal intensity. To account for intensity fluctuations resulting from media switching, a 5-time-point smoothing was performed and the signals at indicated time points (i.e., Supplementary Fig. 7e) were normalized and plotted. It should be noted that such a long time-lapse imaging would unavoidably introduce phototoxicity, which might contribute to the relatively large fluctuations in the resulting data.

**RT-qPCR and ChIP-qPCR experiments.** For RT-qPCR experiments, monoclonal CHO-Gal4 cells were plated on 6 well plates. The plating density was controlled such that the culture would reach ~80–90% confluency at the time of collection. 12 h post-plating, doxycycline (0.1 µg/mL) and rapamycin (0 or 50 nM) were added to the cell culture. 12 h post-induction, the medium was replaced and doxycycline was removed (and rapamycin concentration was unchanged). Cells were collected every 12 h from 0 to 48 h for RNA extraction. RNA was extracted by using RaPure Total RNA Micro Kit (Magen, R4012-02). For each reaction, $10^6$ cells were used for RNA extraction. We used iScript™ cDNA Synthesis Kit (BIORAD, #1708891) for cDNA synthesis. The cDNA products were quantified by qPCR (GOTAQ qPCR Master Mix, Promega), and the reaction system was 15 µL (2 µL cDNA product, ~50 ng). The following qPCR primers were used:
Primer1_F: tgcgacgatgagccgatccata
Primer1_R: gtgagttcgggaaggttgtcgc
Primer2_F: tgtcggcttcacgatgcgaaa
Primer2_R: ctgttggtgcggcggaagaa
Gapdh_CHO_F: GAAAGCTGTGGCGTGATGG
Gapdh_CHO_R: TACTTGGCAGGTTTCTCCAG
Gnb1_CHO_F: CCATATGTTTCTTTCCCAATGGC
Gnb1_CHO_R: AAGTCGTCGTACCCAGCAAG

For ChIP-qPCR experiments, monoclonal CHO-Gal4 cells were plated on 10 cm culture dishes. The plating density was controlled such that the culture would reach ~80-90% confluency at the time of collection (~$10^7$ cells for each reaction). 12 h post-plating, doxycycline (0.1 µg/mL) and rapamycin (0 or 50 nM) were added to the cell culture. 12 h post-induction, the medium was replaced and doxycycline was removed (and rapamycin concentration was unchanged). Cells were collected after 24 or 96 h for ChIP experiments. We strictly followed the protocol from the SimpleChIP Kit (Cell Signaling, #CST 9003S). The following antibodies were used: Tri-Methyl-Histone H3 (Lys27) (C36B11) Rabbit mAb (CST #9733S, 1:100 dilution), Acetyl-Histone H3 (Lys27) (D5E4) Rabbit mAb (CST #8173, 1:100 dilution). For immunoprecipitation reactions, samples were incubated for 4 h at 4°C with rotation. The DNA products were quantified by qPCR (GOTAQ qPCR Master Mix, Promega), and the reaction system was 15 µL (2 µL DNA product, ~50 ng). The following qPCR primers[71] were used:
Primer1_F: ACGGGATCGCTTTCCTCTGAAC
Primer1_R: GAAACTCGGTACCGACTAGTGGC
Primer2_F: GCCACTAGTCGGTACCGAGTTTC
Primer2_R: TATAGGCCTCCCACCGTACACG
Primer3_F: CGTGTACGGTGGGAGGCCTATA
Primer3_R: ATCGGTCCCGGTGTCTTCTATGGA
Primer4_F: TCCATAGAAGACACCGGGACCGA
Primer4_R: TATGGATCGGCTCATCGTCGCA
Bglap2_F: CTAATTGGGGGTCATGTGCT
Bglap2_R: CTTATAAAAGACTGGCTCCAGC
Cdx2_F: GTCTCCAGCCATTGGTGTCT
Cdx2_R: GTCTCCAGCCATTGGTGTCT

**Simulations of sustained transcriptional response.** In order to validate the proposed mechanism in Supplementary Fig. 9b, we carried out deterministic simulations of two cell populations with identical initial TF level distributions, allowing us to compare the responses of downstream target protein under two different input–output functions. More specifically, the two cell populations (5000 cells each) have an identical normally distributed initial TF levels with a mean of 1500 and a standard deviation of 1000. The production rate of downstream target gene follows the given input–output function. For zero rapamycin condition, the input–output function has a Hill coefficient of 1.19 and a $K_d$ of 60,000 (from Supplementary Fig. 6a). For the condition with 50 nM rapamycin, the Hill coefficient is 1.88 and the $K_d$ is 250 (from Supplementary Fig. 6a). Both TF and target protein degrade with the same degradation rate constant, which corresponds to a half-life of ~12 h (estimated based on Supplementary Fig. 7b). For the initial target protein level, we used the steady-state level calculated by the Hill function as an approximation. mRNA was assumed to be at steady state and was not considered

in our model. Based on the initial conditions, we used the following ODE equations to simulate the temporal trajectories of both the TF and target protein levels:

$$\frac{d[TF]}{dt} = -\alpha * [TF] \tag{2}$$

$$\frac{d[Target]}{dt} = \frac{\beta * [TF]^n}{K_d{}^n + [TF]^n} - \alpha * [Target] \tag{3}$$

Here, α represents degradation rate constant of TF and target protein, $n$ represents the Hill coefficient, $K_d$ represents the dissociation constant, β represents the maximal rate of target protein expression when the TF is saturated.

**Reporting summary.** Further information on research design is available in the Nature Research Reporting Summary linked to this article.

## Data availability
Source data are provided with this paper. All other data are available from the corresponding author on request.

## Code availability
All relevant codes are available from the corresponding author on request.

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

## Acknowledgements

This work was supported by grants from National Key R&D Program of China (Grant No. 2020YFA0906900 (Y. Lin), 2018YFA0900703 (Y. Lin)) and National Natural Science Foundation of China (Grant No. 32088101 (Y. Lin), 31771425 (Y. Lin)). We thank the flow cytometry core at the National Center for Protein Sciences at Peking University and the Quantitative Imaging facility at the Center for Quantitative Biology at Peking University for equipment supports.

## Author contributions

J.W. and Y. Lin conceived and designed the study; J.W. and B.C. performed the experiments and analyzed the data, with inputs from Y. Lin; Y. Liu, L.M. and W.H. contributed experimental materials and provided help to the experiments; J.W., B.C. and Y. Lin wrote the paper, with inputs from all authors.

## Competing interests

J.W. and Y. Lin submitted a patent application (Chinese Patent Application No. 2021107851921) on the construction and application of a chemically controlled gene regulatory device based on the technique developed in this paper. All other authors declare no competing interests.
