## [Peer Review File · Nature Communications]

Reviewers' Comments:

Reviewer #1:

Remarks to the Author:

Review: Nature Communications

Transcriptional condensates have been gaining attention as they might play a significant role in gene regulation. The manuscript by Wu et al. describes control of clustering behavior of synthetically constructed transcription factors (TF). The authors quantify TF properties such as gene expression levels and ability of TF clustering propensity to modulate gene transcriptional dynamics as response to chemical stimuli. As the authors pointed out, their previous study (Ma et al., Molecular Cell, 2021) explored transcriptional activation of very similar systems during transient transfection. What distinguishes the work described here is stable transfection of cell lines, multiple cell lines and detailed analysis of transcriptional clustering. This study could provide guidance in the literature that can help others to study transcriptional dynamics. I have some comments regarding characterization of the systems and explanations of mechanisms. In the end, I am suggesting publication with major revision after the following comments have been addressed. We also recommend the removal of the claim of "memory like" as discussed in our comment on Figure 5.

Major:

- Page2 bottom; What is the exact meaning of "clustering propensity"? Does it indicate the size, concentration, and the number of clusters? The authors need to provide the exact definition for better understanding.
- Figure 1a; The author used GFP for both clustering mediator part and synthetic TF part. To confirm the binding of FKBP and FRB under rapamycin, it would be recommended to use different colors for each part, for example, GFP for synthetic part and mCherry for clustering mediator part. It may help to find reasons why TF clusters even exist without rapamycin.
- Figure 1c and e; Is the average number of TF clusters in Figure 1e in a single z plane? It is confusing. For example, the representative images with 500nM rapamycin in Figure 1c show over 20 clusters in a single z plane. It would be better to provide more details on the imaging method.
- Figure 1; Would it be possible for the authors to address any shortcomings in detection/counting the number of clusters in response to rapamycin concentration? Pictures and numbers do not seem to exactly match.
- Could the authors explain what exactly they mean with 'leaky' explanation?
- Figure 3; The authors show 4% of TF clusters are colocalized with PCP foci at 100 nM rapamycin with doxycycline. According to Figure 1e with 100 nM rapamycin, 5 TF clusters on average are in a single cell. It means that only one cell among five cells has one colocalization. The authors argued that they show the direct evidence of TF clustering effect on transcription, but it is hard for me to agree that the 50-fold change in reporter transcription was dominated by this one colocalization among five cells. Of course, as the authors pointed out, clustering might occur but not be captured by microscope. I agree with this statement, but I suggest detailed analysis to address it. Firstly, how much percent are the PCP foci colocalized with TF clusters? ($\#$ of colocalization / total $\#$ of PCP foci) Also, to confirm the locations and the number of reporter genes, DNA-FISH for PP7 is recommended. Then the authors can analyze the colocalizations of TF clusters and the reporter gene loci. It helps to understand how many reporter genes are on activation during imaging time and the dynamics between TF clusters and reporter genes.
- Figure 5; It describes a telegraph model for input-output relationship. The authors cannot claim memory as memory would suggest that subsequent (future) events would be directly dependent on the prior events (here the authors only test one event). The authors should remove the memory claim, and the Figure 5 can be a supplementary figure.

Minor:

- What is the exact meaning of "clustering propensity"? Does it indicate the size, concentration, and the number of clusters? The authors should provide the exact definition for better understanding.
- Page3 bottom; Is it a typo? (tetrameter -> tetramer)
- Page3 bottom; This paragraph does not include the description of the "Clustering Mediator" part.
- All figures; The authors should put error bars, for example, Figure 1 c,d, e and Figure 4 d, e do not contain error bars.
- Page4 middle; The author says the mean cluster diameter saturates at high concentration of

rapamycin. But the Figure 1e-f shows the mean cluster diameter keeps increasing while the mean cluster number saturates.

- Fig S1 part a; Does a point in TF concentration mean the average GFP intensity of all clusters in the cell? It would be better to add more details on how the authors measure the TF concentration. What is the criterion of arbitrary units?

- Fig S1 part c; for consistency (Frb -> FRB)

- Figure 2e; The authors show CHO-w/oTAD as a control to test the effect of trans-activation domain(TAD). This result shows a rigid conclusion. However, most experiments are performed with U2OS-7TetO cell line, so it would be better if the authors can show a control experiment using U2OS-7TetO cell line lacking TAD.

- Figure 2e; It would be easy to understand if its y-scale shows 1. (For example, 0, 0.2, 0.4, 0.6, 0.8 and 1.0) Also, I think sharing the same y-scale for each figure can keep its consistency. For example, if Figure 2d and 2e have the same y-scale, it would be easier to understand (you can add inner figure to highlight the details for 2e)

- Fig S1d, Fig S3a,b; What is the meaning of "density"? And the unit for this density is quite ambiguous. Is it a normalized distribution?

- Figure 4b; It needs axes properly labelled.

- Figure 5d, Fig S5b; Figure 5d is from gated cells and Fig S5b is from ungated cells. But their distribution in iRFP intensity looks identical for me. To check this is not coincidence, it would be better for the authors to provide iRFP intensity distribution of only-ungated cells as supplementary information.

Reviewer #2:

Remarks to the Author:

In this work by Wu et al, the authors aim to explore whether clustering of transcription factors (TFs) can lead to changes in transcription regulation. This is a timely subject as many studies have suggested a role for clustering/condensates in transcription regulation, but how clusters form and how they regulate transcription remains hotly debated. To address these questions, the authors have developed a chemically-induced system which allows them to titrate the degree of TF clustering by adding increasing concentrations of rapamycin and monitor its effects in gene response. They observe increased levels of gene activation upon induced clustering, which is in nice agreement with recent work by Wei et al. 2020 and Schneider et al. 2021 where light-inducible approaches were employed. The authors further demonstrate novel "memory-like" effects in expression and claim that clustering leads to ultrasensitive responses of cells to intracellular TF concentration. While the authors' developed a nice system and performed careful experiments and the findings are intriguing, two main concerns about the interpretations dampen enthusiasm: first, it is unclear (and seems unlikely) that the macroscopic clusters detected by the authors are actually driving the transcriptional response; second, the changes in the cell response to induced clustering seems mainly driven by a change in sensitivity (Kd) rather than ultrasensitivity (Hill coefficient), which is also an interesting finding. Based on these concerns and others detailed below, we recommend the authors to revise their manuscript significantly.

Major points:

- 1) the fact that transcription activity does not scale with the number or size of the clusters raises a crucial question as to how clustering impacts transcription: do the visible clusters quantified by the authors drive the observed cellular response? At first glance the colocalization of clusters with sites of active transcription seems too low to explain the robust transcriptional response observed: at 100 nM rapamycin, cells typically exhibit 4 TF clusters (Figure S1A), which translates into 0.16 colocalized TF-transcription sites per cell (based on the ~4% frequency of observing a transcription site near a TF cluster, Fig. 3c). So ~1 in 6 cells is expected to show a visible cluster present at a transcription site; since each cell typically exhibits ~10 transcription sites at that rapamycin concentration, it seems that TF cluster-induced transcription would only explain ~1.6% of observed transcription? If this reasoning is correct, it seems unlikely that the visible TF clusters drive the effect; rather, clustering below the detection threshold would be the likely driver. To clarify the possible role of sub-detection level TF clusters at or near sites of transcription, it would be important to measure TF cluster-transcription activity correlations at the allele level, e.g. detect all transcription sites, bin them by increasing PCP intensity levels and quantify the presence, size

and brightness of any associated cluster for the different PCP intensity bins (this is conceptually similar to the analyses in Figures 3c and S2, but using transcription sites as masks rather than TF clusters). If the large clusters indeed drive transcription, one would expect that highly active transcription sites are more likely to associate with large/bright clusters. Alternatively, there might be an optimal clustering regime for transcription response, which would be a novel result. This experiment would help address whether large or smaller clusters are driving the observed effect and/or whether large clusters are just passengers.

2) In figure 4, the change in K_d (multiple orders of magnitude) seems the major driver of the effect compared to a more modest contribution from changes in the Hill coefficient. This raises the question whether clustering drives ultrasensitivity as claimed by the authors, or rather modulates the sensitivity range of the system via K_d changes. The robustness of the Hill coefficient result is also not totally clear, since the lower rapamycin concentrations only sample a small fraction of the Hill curve and therefore the Hill coefficient estimates could be biased for some conditions: at 0.1 nM rapamycin, TF concentrations seem to reach at most $\sim 30,000$ AU, almost an order of magnitude below the fitted K_d of $\sim 200,000$ AU.

The authors should fit all the K_d on their curves while keeping the Hill coefficient fixed (e.g. using an average of the K_d s from their current fits), and then compare the fit quality of the fixed Hill coefficient results with their current fit results; they should use a statistical metric such as the Akaike information criterion to decide whether the increase in goodness of fit justifies adding an extra parameter. If it turns out the clustering modulates the sensitivity range of the system via K_d changes (which is also interesting!), the text should be rewritten accordingly, e.g. claims that "the nonlinearity in the system becomes larger when the TF clustering propensity increases" should be modified.

3) In the memory experiment, rapamycin-specific effects on translation and protein degradation could impact iRFP readout by modulating iRFP translation or stability. The authors should control that transcription activity tracks with the observed trend at the protein level, either using their PCP-based transcription reporter, or by RT-PCR.

Minor points:

- Figure 3C/S2B: are there fewer clusters in the cells when there is no Dox which would skew the results? It would be helpful to include a "pseudo clusters" control to the no Dox condition. The expectation would be that there would be no enrichment in clusters in the no Dox above no Dox pseudo clusters.
- Figure 4b: axes are unclear.
- Error bars are needed in Figure 1 d-f, 4 c-e, some supplementary figures
- Figure 5: c-d: $n = ?$
- initiation time is a term that could be confusing in the context of transcription as it often refers to the initiation step of each transcription cycle - "time to first burst" might be a better term to describe the induction of the system?
- In the memory experiment, could the author quantify the visible clusters remaining over time?
- Could the authors quantify the lifetime of the clusters upon puromycin wash-off? This could narrow down models explaining the memory observed, e.g. do clusters remain physically present or do they confer long-lived transcription enhancement beyond their lifetime.
- The authors should clearly define "ultrasensitivity" in the introduction.
- Page 4, second last paragraph - "there are additional parameters such as protein expression level that can affect the clustering propensity at the level of individual cells" - do the authors observe any evidence of this in their system?
- Page 6, second last paragraph - the authors speculate the short dwell time of TF clusters on DNA being one of the reasons for the low % of overlaps between induced clusters and the active transcription sites. Do any of the observed clusters move around over time, and can they measure the dwell time of clusters at transcription sites using their PCP reporter?

Dear Reviewers,

We greatly appreciate your time and the constructive comments and suggestions, which have truly helped us to improve the manuscript. In response to your comments and suggestions, we have performed a series of new experiments and analyses, and have obtained significant new evidence supporting gene regulatory roles of rapamycin-induced TF clusters. We believe that, with these new results, our work has convincingly established the DNA locus-specific binding capability and transcriptional activation capability of TF clusters, and has provided important insights into the emergent gene regulatory functions conferred by TF clustering.

In the revised manuscript, we have incorporated these new results, and have largely improved the text and figures to comprehensively address all your comments. These are described in the point-by-point reply. Here we first summarize the major changes in the revision:

- We have performed DNA FISH assay to establish the locus-specific binding of TF clusters. Importantly, we showed that the DNA binding capability of TF clusters is doxycycline dependent.
- We further preformed two-color time-lapse imaging assay to show that TF clusters can spatiotemporally interact with nascent transcription sites. Together with DNA FISH assay, new results from this experiment establish the locus-specific gene regulatory function (i.e., DNA binding and transcriptional activation) of TF clusters.
- We have also carefully addressed the concern regarding Hill function fitting. Importantly, we carried out local sensitivity analysis of the dose response data (without Hill fitting) to demonstrate the modulation of local sensitivity by TF clusters.

Additionally, we would like to note that the title of the manuscript was slightly altered in response to the comment that “memory” claim could be confusing (please refer to the response below on page 9). We have also uploaded a version of the text with Track Changes to highlight the edits made to the original text.

Below please find a summary of changes to the figures:

Main figures:

1. **Figure 1a-b:** minor corrections to the original schematic figure were made.
2. **Figure 1c:** replaced several images with new ones to better represent the mean behavior of cells of the corresponding condition.
3. **Figure 1d-f:** original code was modified to further improve cluster detection and the plots were updated with shading to indicate 95% CI. We now show example outputs of the code in Supplementary Fig. 1a to demonstrate the performance of algorithmic cluster detection.
4. **Figure 2b:** included new data from a control cell line, i.e., U2OS-w/oTAD. The presentation of all data has been revised to improve clarity.
5. **Figure 3a:** schematic figure was modified to improve clarity.
6. **Figure 3b:** a new figure panel based on results of DNA FISH assay showing TF clusters can directly bind to target gene loci.

7. **Figure 3c:** a new figure panel to quantify the co-localization events per cell between TF clusters and gene loci in both dox and w/o dox conditions.
8. **Figure 3d:** a new figure panel showing two-color time-lapse snapshots of TF clusters and nascent transcription sites.
9. **Figure 3e:** a new figure panel showing time-lapse trace of fluorescence intensities of indicated TF clusters and nascent transcription site.
10. **Figure 3g:** new analysis result was included and code for the calculation of co-localization fraction between TF clusters and PCP signals in control condition was slightly modified.
11. **Figure 4a:** schematic figure was modified to improve clarity.
12. **Figure 4b:** new experimental data was collected and analyzed; three representative rapamycin concentrations were selected for comparison.
13. **Figure 4c:** new experimental data was collected and analyzed; the plot was updated.
14. **Figure 4d:** a new figure panel showing the modulations of ultrasensitivity and effective binding affinity by TF clustering in CHO cells. The original plot was moved to the supplement.
15. **Figure 4e:** a new figure panel showing TF clustering can contribute to ultrasensitive input-output function independent of tandem binding sites. The original plot was moved to the supplement.

Supplementary figures:

1. **Supplementary Fig. 1a:** a new figure panel showing the result of the algorithm for automatic TF cluster detection.
2. **Supplementary Fig. 1b-c:** original code was modified to improve TF cluster detection and the plots were updated.
3. **Supplementary Fig. 1d:** a new figure panel demonstrating rapamycin induced co-clustering of synthetic TF and clustering mediator.
4. **Supplementary Fig. 1e:** minor corrections to the original schematic figure were made.
5. **Supplementary Fig. 2a:** a new figure panel showing that TF clusters can bind to target genes in a dox-dependent manner.
6. **Supplementary Fig. 2b:** a new figure panel showing the number of detected DNA FISH loci per cell.
7. **Supplementary Fig. 2c:** a new figure panel showing the fraction of FISH loci co-localizing with TF clusters per cell.
8. **Supplementary Fig. 2d:** a new figure panel showing that the largest TF clusters might not possess the strongest DNA binding capability.
9. **Supplementary Fig. 3a:** a new figure panel showing the spatiotemporal interactions between TF clusters and nascent transcription sites using two-color time-lapse imaging.
10. **Supplementary Fig. 3b:** a new figure panel showing the dwell time of TF clusters binding to the reporter locus.
11. **Supplementary Fig. 3c:** a new figure panel showing several potential interactions modes between TF clusters and nascent transcription sites.
12. **Supplementary Fig. 3d:** code for the calculation of co-localization fraction of TF clusters and PCP signals in control condition was slightly modified, and the plot was updated.
13. **Supplementary Fig. 3f-g:** new figure panels showing a much higher overlap rate with TF

clusters for real PCP signals compared to pseudo-PCP signals.

14. **Supplementary Fig. 3h:** a new figure panel showing a negative correlation between the intensities of TF clusters and nascent transcription sites.
15. **Supplementary Fig. 5a-b:** new experimental data was collected and the plots were updated.
16. **Supplementary Fig. 5c:** new experimental data was collected and the error bar for bimodality index was added.
17. **Supplementary Fig. 5d:** new experimental data was collected and analyzed, and for all rapamycin concentrations, one replicate was selected for comparison.
18. **Supplementary Fig. 5e:** new experimental data was collected and the error bars for Hill coefficient and Kd were added.
19. **Supplementary Fig. 5g:** a new figure panel showing the squared norm of the residual reaches minimal at varying Hill coefficients under different rapamycin concentrations.
20. **Supplementary Fig. 5h:** a new figure panel showing the local sensitivity increases as rapamycin concentration increases.
21. **Supplementary Fig. 5i-j:** new experimental data was collected and the error bars for standard deviation and coefficient of variation were added.
22. **Supplementary Fig. 6a:** code for Hill function fitting was slight modified and figures were updated (analogous to Figure. 4d).
23. **Supplementary Fig. 6b-c:** code for Hill function fitting was slightly modified, the Hill coefficient value and Kd were updated and the error bar was added.
24. **Supplementary Fig. 6d:** added the error bar of bimodality index.
25. **Supplementary Fig. 6e:** a new figure panel showing that local sensitivity analysis on the CHO-Gal4 data.
26. **Supplementary Fig. 6g:** added the error bar for Hill coefficient.
27. **Supplementary Fig. 6h:** added the error bar for Kd.
28. **Supplementary Fig. 6i:** added the error bar for bimodality index.
29. **Supplementary Fig. 7a:** a new figure panel showing the gated and ungated flow cytometry data from the experimental time course.
30. **Supplementary Fig. 7d:** a new figure panel showing the mRNA level decays much slower at high clustering propensity compared to the no rapamycin condition.
31. **Supplementary Fig. 7h:** a new figure panel showing the number of visible TF clusters decreased over the time course.
32. **Supplementary Fig. 9d:** new simulation was conducted based on new fitted values.

Reviewer #1 (Remarks to the Author):

Review: Nature Communications

Transcriptional condensates have been gaining attention as they might play a significant role in gene regulation. The manuscript by Wu et al. describes control of clustering behavior of synthetically constructed transcription factors (TF). The authors quantify TF properties such as gene expression levels and ability of TF clustering propensity to modulate gene transcriptional dynamics as response to chemical stimuli. As the authors pointed out, their previous study (Ma et al., *Molecular Cell*, 2021) explored transcriptional activation of very similar systems during transient transfection. What distinguishes the work described here is stable transfection of cell lines, multiple cell lines and detailed analysis of transcriptional clustering. This study could provide guidance in the literature that can help others to study transcriptional dynamics. I have some comments regarding characterization of the systems and explanations of mechanisms. In the end, I am suggesting publication with major revision after the following comments have been addressed. We also recommend the removal of the claim of “memory like” as discussed in our comment on Figure 5.

We greatly appreciate the very helpful comments and suggestions. Based on the referee’s comments and suggestions, we have performed a series of new experiments and analyses, including analysis of new cell lines, two-color time-lapse imaging, and DNA FISH imaging of target gene loci, and have substantially revised the manuscript to address the comments and concerns raised by the referee. In the following we provide a detailed point-by-point response to the comments.

Major:

- Page2 bottom; What is the exact meaning of “clustering propensity”? Does it indicate the size, concentration, and the number of clusters? The authors need to provide the exact definition for better understanding.

We thank the referee for raising an important issue regarding the terminology. Regarding the exact meaning of “clustering propensity”, we meant to use the term to describe the observation that the overall degree of clustering in our system increases as rapamycin increases, whereby the degree of clustering can be quantified by either the fraction of cells with visible clusters, the number of clusters per cell, or the mean size of the cluster. Importantly, compared to other terms such as clustering strength, cluster size, and cluster number, we felt that clustering propensity better describes the apparently probabilistic nature of cluster formation in our system, as rapamycin alters the probability of cells forming visible clusters.

We have now included additional text to clarify the definition of clustering propensity and discuss the rationale for using this specific term (**Line 141-144** in the main text). Note that in the results section, we now avoid the usage of “clustering propensity” before the definition of this term.

- Figure 1a; The author used GFP for both clustering mediator part and synthetic TF part. To confirm the binding of FKBP and FRB under rapamycin, it would be recommended to use different colors for each part, for example, GFP for synthetic part and mCherry for clustering mediator part. It may help to find reasons why TF clusters even exist without rapamycin.

We thank the referee for suggesting a very useful experiment. As the referee noted, this experiment would serve two purposes: 1) to confirm that rapamycin is indeed responsible for the induced binding FKBP and FRB (and thus the induced formation of TF clusters); and 2) to delineate the potential reasons for why TF clusters even exist in the absence of rapamycin. Following the referee's suggestion, we have carried out new experiments which provide insights into the above two aspects. More specifically,

- We constructed a new construct where the synthetic TF and the clustering mediator are separately fused with GFP and mCherry. We then transfected this construct into U2OS cells, and found that the two FPs (that are fused to synthetic TF or the cluster mediator) can be induced to form co-localized clusters by rapamycin, consistent with the expected behavior of the system. In the absence of rapamycin, we found that each of the two components can separately form cluster-like fluorescence signals, and these clusters are non-overlapping between the two colors (**Supplementary Fig. 1d**).
- Based on these results and the result showing that target reporter was barely activated without rapamycin, we reason that the observed TF clusters in the absence of rapamycin (in the single-color cell line) likely resulted from either one component instead of from their co-clustering, and that these clusters possess a much weaker gene activation capability compared to the co-clusters.

Together, these results validate the design of our inducible clustering system and provide new insights into the behavior of the system. We have now included these new results in the manuscript (**Line 152-163** in the main text).

- Figure 1c and e; Is the average number of TF clusters in Figure 1e in a single z plane? It is confusing. For example, the representative images with 500nM rapamycin in Figure 1c show over 20 clusters in a single z plane. It would be better to provide more details on the imaging method.

We thank the referee for suggesting further clarifications on the images and imaging method. We apologize for not including the imaging method in the original figure caption. Each image in Fig. 1c was a maximum intensity projection of 9 confocal z-slices, and the imaging method was described in the Methods section. We now include the information of the imaging method in the figure caption to improve clarity (**Line 983** in the main text). We have also replaced images with ones having cluster numbers close to the mean numbers of the corresponding condition.

- Figure 1; Would it be possible for the authors to address any shortcomings in detection/counting the number of clusters in response to rapamycin concentration? Pictures and numbers do not seem to exactly match.

We thank the referee for suggesting further clarifications on the quantification of images. We realized that it was indeed confusing that the numbers of clusters in the images did not match the summarized statistics. We have done several things to address the concern raised. More specifically,

- In Fig. 1E, the mean cluster number over cells was shown, while in Fig. 1C some cells contain more than the mean cluster number. We realized that such a mismatch is indeed confusing, and we have replaced some cells in Fig. 1C with ones that behaved closer to the mean. It should be noted that in Supplementary Fig. 1b-c we presented dot plots showing individual cell behavior as well as boxplots showing the population statistics.
- We now show example images of cells with different cluster numbers under the same rapamycin (**Supplementary Fig. 1a**). In these images, we overlaid detected clusters by our algorithm, which helps to illustrate the definition and quantification of TF clusters (in addition to descriptions in the Methods section).

Additionally, we have added more discussion on the cluster detection and analysis in the Method section (**Line 592-594** in the main text), and have included additional notes on the cell-to-cell variability of cluster properties in the results section (**Line 149-151** in the main text). We think these have provided a clearer picture on the analysis of clusters and the heterogeneity among cells under the same rapamycin.

- Could the authors explain what exactly they mean with ‘leaky’ explanation?

We thank the referee for suggesting further clarification on the term “leaky”. In the original Methods section, we used “leakiness” to describe the potential reasons for the appearance of clusters without rapamycin. Because “leakiness” is often used in synthetic biology to describe basal expression of synthetic genes, which is uncontrolled and often due to enhancer interference from adjacent genes, we thus used “leakiness” to describe the uncontrolled clustering of TFs. Based on the new experiment showing that the two components can separately form clusters, we now have removed this term (and the associated text) and have explained more specifically how clusters could form in the absence of rapamycin (**Line 163-165** in the main text).

- Figure 3; The authors show 4% of TF clusters are colocalized with PCP foci at 100 nM rapamycin with doxycycline. According to Figure 1e with 100 nM rapamycin, 5 TF clusters on average are in a single cell. It means that only one cell among five cells has one colocalization.

The authors argued that they show the direct evidence of TF clustering effect on transcription, but it is hard for me to agree that the 50-fold change in reporter transcription was dominated by this one colocalization among five cells. Of course, as the authors pointed out, clustering might occur but not be captured by microscope. I agree with this statement, but I suggest detailed analysis to address it. Firstly, how much percent are the PCP foci colocalized with TF clusters? ($\#$ of colocalization / total $\#$ of PCP foci) Also, to confirm the locations and the number of reporter genes, DNA-FISH for PP7 is recommended. Then the authors can analyze the colocalizations of TF clusters and the reporter gene loci. It helps to understand how many reporter genes are on activation during imaging time and the dynamics between TF clusters and reporter genes.

We thank the referee for raising important issues regarding the interpretation of the colocalization experiment and the mechanism of transcriptional activation by TF clusters. To address these issues, we have conducted additional analyses, performed two-color time-lapse imaging experiments to analyze the spatiotemporal interactions between TF clusters and gene transcription, and carried out the DNA FISH experiment where we were able to simultaneously imaged TF clusters and PP7 gene loci. The new results have much clarified the potential function of TF clusters. More specifically,

- We would like to first discuss whether the 4 % overlap (i.e., 4% of TF foci colocalizing with PCP signals) could relate to gene expression enhancement by TF clusters, and outline new experiments (which we have already performed) to better establish the causality between TF clusters and transcriptional activation.
- We first performed the suggested analysis to calculate the fraction of nascent transcription sites overlapping with TF clusters using the same cells as in Fig. 3g. We found that about 10% of the PCP-mCherry foci exhibited co-localization with TF clusters (**Supplementary Fig. 3f-g**). This co-localization rate is consistent with the \sim 4% overlap rate when the denominator is the number of TF clusters. More specifically, on average there are 6.4 TF clusters and 3.5 PCP-mCherry foci per cell. In about 3.9 cells, there would be 1 overlap event based on 4% overlap rate, corresponding to 1 in 13.6 or \sim 7.3% PCP-mCherry foci. This result indicates that indeed there was a relatively low (but statistically much higher than random) overlap rate between TF clusters and nascent transcription sites when analyzing static snapshots of hundreds of cells.
- Regarding the 4% overlap (or 10% overlap when dividing by PCP foci number), we understand that it is challenging to relate this small percentage of overlap to gene expression enhancement by TF clusters, as the referee already articulated above. In fact, we did not attempt to make such a quantitative connection between co-localization and transcriptional enhancement in the original manuscript. Rather, we emphasized on the important implication from the 4% overlap, i.e., the TF clusters are likely functional for gene activation. More specifically, the claim that TF clusters are likely casual for gene activation was based on two parallel observations (in addition to evidence shown in Fig.2): a) without doxycycline, TF clusters showed background-level co-localization with

PCP speckles, which are likely not real transcription signals as the target expression is indistinguishable from background (Supplementary Fig. 1f); b) with doxycycline, TF clusters were able to bind to DNA and we observed clear transcription signals and co-localization between such signals with TF clusters.

- We agree with the referee that the claim that TF clusters are activating transcription should be further supported, and additional experiments should be done. Along the line of the suggested experiments, we think two sets of new experiments are needed. First, if TF clusters are indeed involved in activating gene transcription, analyzing how TF clusters spatially and *temporally* interact with nascent transcription sites would be highly desirable, as such data could inform whether TF clusters are stably or transiently bound to the gene loci, and how such binding correlates with transcriptional activity. Second, because previous experiments only examined the interactions between TF clusters and nascent transcription sites, it would be necessary to provide a direct evidence for the binding of TF clusters to the chromosomal loci of the reporter genes. We thus need to simultaneously image DNA (e.g., using DNA FISH) and TF clusters, as such data would not only inform whether TF clusters bind to target sites, but also allow comparing the fractions of TF clusters co-localizing with DNA versus with nascent RNAs.
- We next describe the results and implications of the new experiments and analyses.
 - To capture spatiotemporal interactions between TF clusters and nascent transcription sites, we carried out two-color (GFP and mCherry) time-lapse imaging of the U2OS-7TetO cell line at a relatively high temporal resolution (5 min per frame) over >20 hours. We observed frequent events where TF clusters and nascent transcription sites temporally overlap (**Fig. 3d-e, Supplementary Fig. 3a-c, Supplementary Movie 1**). Intriguingly, TF clusters appeared to interact with nascent transcription sites in various ways (**Supplementary Fig. 3c**). For example, there are examples where TF clusters appeared to repeatedly activate reporter transcription, and there are also examples where TF clusters exhibited a “hit-and-run” behavior that leads to sustained reporter activation. There are also scenarios where TF clusters could not be visually identified surrounding the transcription sites (i.e., TF cluster-independent activation). These new results, together with the evidence of significant co-localization from the snapshots, strongly indicate the causal role of TF clusters in the transcriptional activation of target genes.
 - We further carried out DNA-FISH assay to image the reporter loci (labeled with Cy5) in the genome (labeled with DAPI). In the same cell, we were able to acquire GFP signals, allowing us to analyze the co-localization between TF clusters and reporter gene loci in fixed cells (**Fig. 3b-c, Supplementary Fig. 2**). From the acquired three-color images (Cy3, DAPI, and GFP), we found that cells often contained TF clusters co-localizing with reporter gene loci in the presence of doxycycline (15 out of 16 representative cells). In contrast, in the absence of doxycycline, such co-localization was rarely observed (1 out of 16 representative

cells). These results strongly suggest that TF clusters can indeed specifically bind to reporter gene loci when the DNA binding capability is switched on.

- One intriguing observation from the above DNA-FISH experiment was that the fraction of chromosomal gene loci co-localizing with TF clusters (~15%) is higher than the fraction of nascent transcription sites overlapping with TF clusters (~10%). One potential explanation is that for each gene locus activated by TF cluster, when TF cluster is bound to the DNA, the fraction of time transcription site appears is ~67%.
- Using two-color dynamic traces of individual gene locus measured from the time-lapse movies, we could estimate the fraction of time that transcription signal appeared when TF cluster was observed at a specific gene locus. Based on two-color traces we found that when GFP signal was detected, 66.8% of the time mCherry (PCP) signal was also detected, coinciding with the 67% estimate from the two static measurements (i.e., snapshots of DNA-FISH and static snapshots of TF and PCP signals). It is reassuring that there is a nice agreement between the two estimates based on completely different measurement modalities
- It is noted that these new results have led to a significant revision of the manuscript. More specifically, we now first present the data from DNA FISH assay as the evidence supporting the locus-specific DNA binding ability of TF clusters (**Line 233-251** in the main text, **Fig. 3b-c, Supplementary Fig. 2**). We then present the data from the two-color time-lapse imaging experiment as the evidence supporting the locus-specific transcriptional activation capability of TF clusters (**Line 253-268** in the main text, **Fig. 3d-e, Supplementary Fig. 3a-c, Supplementary Movie 1**). We next use the original snapshot data as the evidence supporting the DNA-binding dependent, locus-specific transcriptional activation of target genes by TF clusters (**Line 270-290** in the main text). The discussion section has also been revised accordingly (**Line 473-483** in the main text) .

Together, we believe that these new experiments and analyses have provided much stronger supports for the gene regulatory role of TF clusters, and have thus substantially improved the revised manuscript.

- Figure 5; It describes a telegraph model for input-output relationship. The authors cannot claim memory as memory would suggest that subsequent (future) events would be directly dependent on the prior events (here the authors only test one event). The authors should remove the memory claim, and the Figure 5 can be a supplementary figure.

We thank the referee for raising an important concern regarding our claim of memory. We would like to first explain the rationale for the memory claim, and describe what we have done in the revised manuscript to alleviate potential confusions and to improve clarity.

- We note that the referee pointed out one definition of memory, where the prior

experience (such as experiencing a transient stress) of a system influences the subsequent response of the system. This definition has been used to determine the presence or absence of memory in transcriptional regulation (i.e., transcriptional memory). Transcriptional memory is often associated with epigenetic mechanisms, and a notable example is the lipopolysaccharide-induced transcriptional memory in macrophages (Foster, Nature 2007).

- We would like to point out another definition of memory, in which a transient stimulus elicits a long-lasting response. This definition of memory is similar to the previous definition in that they both emphasize the long-lasting nature of the response, but differs in that the second definition relies on quantifying whether the response is indeed long-lasting compared to the stimulus duration, instead of using a subsequent stimulus to test the presence of memory as in the previous definition. This second definition of memory has been widely used in systems and synthetic biology fields to describe circuits that have the ability to elicit a sustained response to transient inputs. A classic example is the synthetic toggle switch in *E. coli* (Gardner, Nature 2000), where a transient inducer input can evoke a long-lasting, irreversible switch in the reporter state. Note that in this example, a second inducer input was not necessary to test the presence of memory.
- In our system, we quantitatively determined that a transient input can indeed elicit a relatively sustained response of the target gene, and such behavior depends on the presence of rapamycin. Based on these results and the second definition of memory, we claimed that our system exhibited memory-like behavior.
- Nevertheless, because the memory claim could cause potential confusions to some readers, we now describe the phenomenon as it is, i.e., sustained transcriptional response to transient inputs. We regard this as an important result of the manuscript because it indicates the potential of TF clusters eliciting a cell state memory during cell fate transition, and thus would like to keep Fig. 5 as a main figure. As such, we have now removed the memory claim from the manuscript title and the figure title. The manuscript title now becomes “Ultrasensitivity and sustained transcriptional response with chemically controlled transcription factor clustering”. The title of Fig. 5 now becomes “TF clustering confers sustained transcriptional response to transient stimuli”.

In sum, we believe these modifications have much clarified the role of TF clusters in eliciting sustained transcriptional response, and have mitigated potential confusions by avoiding describing the phenomenon as memory (note that we still mention “memory” as an analogy a couple of times in the revised text).

Minor:

- What is the exact meaning of “clustering propensity”? Does it indicate the size, concentration, and the number of clusters? The authors should provide the exact definition for better understanding.

We note that this concern is the same as the first point of the major concerns, which we have addressed above.

- Page3 bottom; Is it a typo? (tetrameter -> tetramer)

Yes, it is a typo. Thank you for catching it – it is now corrected. In the revised manuscript, we have made sure to more carefully proof read the manuscript to avoid mistakes like this.

- Page3 bottom; This paragraph does not include the description of the “Clustering Mediator” part.

We thank the referee for the suggestion. We have now included a detailed description about the “clustering mediator” (**Line 109-111** in the main text).

- All figures; The authors should put error bars, for example, Figure 1 c,d, e and Figure 4 d, e do not contain error bars.

Thank you. We have now included error bars as suggested. All figures have appropriate error bars indicating S.D. or 95% confidence interval.

- Page4 middle; The author says the mean cluster diameter saturates at high concentration of rapamycin. But the Figure 1e-f shows the mean cluster diameter keeps increasing while the mean cluster number saturates.

We thank the referee for pointing out the lack of accuracy in our statement. We have now revised the statement as “the curve for mean cluster diameter has not yet reached an apparent saturation at the maximum rapamycin concentration, while other two curves appear to have saturated at high rapamycin concentrations, indicating that TF clusters tend to grow in size instead of in number beyond a threshold rapamycin concentration at ~100 nM (**Fig. 1e-f**)”. See **Line 138-141** in the main text.

- Fig S1 part a; Does a point in TF concentration mean the average GFP intensity of all clusters in the cell? It would be better to add more details on how the authors measure the TF concentration. What is the criterion of arbitrary units?

We apologize for the lack of clarity for Supplementary Fig. 1. We have now included further details in the legend of Supplementary Fig. 1. More specifically, the relevant part of the legend now reads: “Detected numbers of TF clusters and the corresponding TF concentrations (quantified as mean nuclear EGFP intensities) in individual cells (with each dot representing

one cell) are plotted for U2OS-7TetO cells under various rapamycin concentrations”. We note that “arbitrary unit” was used because the TF concentration was quantified by mean nuclear fluorescence intensity.

- Fig S1 part c; for consistency (Frb -> FRB)

Thank you. We have made the correction.

- Figure 2e; The authors show CHO-w/oTAD as a control to test the effect of trans-activation domain(TAD). This result shows a rigid conclusion. However, most experiments are performed with U2OS-7TetO cell line, so it would be better if the authors can show a control experiment using U2OS-7TetO cell line lacking TAD.

We thank the referee for suggesting an important control experiment. We have collected new data using the newly constructed control U2OS cell line, which contains synthetic TFs without TAD. Reassuringly, the results are as expected, i.e., TAD, but not other components of the system, is responsible for transcriptional activation. Please refer to **Fig. 2b** third panel for the data of this control cell line.

- Figure 2e; It would be easy to understand if its y-scale shows 1. (For example, 0, 0.2, 0.4, 0.6, 0.8 and 1.0) Also, I think sharing the same y-scale for each figure can keep its consistency. For example, if Figure 2d and 2e have the same y-scale, it would be easier to understand (you can add inner figure to highlight the details for 2e)

We thank the referee for suggesting how to improve Fig. 2e. We have now made necessary corrections to ensure that the scale has been properly displayed, which should help deliver the message more clearly. Please refer to Fig. 2b for the revised presentation of the data.

- Fig S1d, Fig S3a,b; What is the meaning of “density”? And the unit for this density is quite ambiguous. Is it a normalized distribution?

Density refers to probability density, and the plot was generated by R using the density plot function. We have now revised the y-axis label to indicate “Probability density” in Supplementary Fig. 5a-b and Supplementary Fig. 1f.

- Figure 4b; It needs axes properly labelled.

We have now revised the axis labels.

- Figure 5d, Fig S5b; Figure 5d is from gated cells and Fig S5b is from ungated cells. But their

distribution in iRFP intensity looks identical for me. To check this is not coincidence, it would be better for the authors to provide iRFP intensity distribution of only-ungated cells as supplementary information.

It is noted that indeed there is only a minor difference in iRFP distributions between gated and ungated samples. We have now plotted the distributions as suggested (see **Supplementary Fig. 7a**).

Reviewer #2 (Remarks to the Author):

In this work by Wu et al, the authors aim to explore whether clustering of transcription factors (TFs) can lead to changes in transcription regulation. This is a timely subject as many studies have suggested a role for clustering/condensates in transcription regulation, but how clusters form and how they regulate transcription remains hotly debated. To address these questions, the authors have developed a chemically-induced system which allows them to titrate the degree of TF clustering by adding increasing concentrations of rapamycin and monitor its effects in gene response. They observe increased levels of gene activation upon induced clustering, which is in nice agreement with recent work by Wei et al. 2020 and Schneider et al. 2021 where light-inducible approaches were employed. The authors further demonstrate novel “memory-like” effects in expression and claim that clustering leads to ultrasensitive responses of cells to intracellular TF concentration. While the authors’ developed a nice system and performed careful experiments and the findings are intriguing, two main concerns about the interpretations dampen enthusiasm: first, it is unclear (and seems unlikely) that the macroscopic clusters detected by the authors are actually driving the transcriptional response; second, the changes in the cell response to induced clustering seems mainly driven by a change in sensitivity (K_d) rather than ultrasensitivity (Hill coefficient), which is also an interesting finding. Based on these concerns and others detailed below, we recommend the authors to revise their manuscript significantly.

We greatly appreciate the very helpful comments and suggestions. Based on the referee’s comments and suggestions, we have performed a series of new experiments and analyses, including DNA FISH assay, two-color time-lapse imaging, qPCR assay, and analyses of Hill fitting, and have substantially revised the manuscript to address the comments and concerns raised by the referee. In the following we provide a detailed point-by-point response to the comments.

Major points:

1) the fact that transcription activity does not scale with the number or size of the clusters raises a crucial question as to how clustering impacts transcription: do the visible clusters quantified by the authors drive the observed cellular response? At first glance the colocalization of clusters with sites of active transcription seems too low to explain the robust transcriptional response observed: at 100 nM rapamycin, cells typically exhibit 4 TF clusters (Figure S1A), which

translates into 0.16 colocalized TF-transcription sites per cell (based on the ~4% frequency of observing a transcription site near a TF cluster, Fig. 3c). So ~1 in 6 cells is expected to show a visible cluster present at a transcription site; since each cell typically exhibits ~10 transcription sites at that rapamycin concentration, it seems that TF cluster-induced transcription would only explain ~1.6% of observed transcription? If this reasoning is correct, it seems unlikely that the visible TF clusters drive the effect; rather, clustering below the detection threshold would be the likely driver. To clarify the possible role of sub-detection level TF clusters at or near sites of transcription, it would be important to measure TF cluster-transcription activity correlations at the allele level, e.g. detect all transcription sites, bin them by increasing PCP intensity levels and quantify the presence, size and brightness of any associated cluster for the different PCP intensity bins (this is conceptually similar to the analyses in Figures 3c and S2, but using transcription sites as masks rather than TF clusters). If the large clusters indeed drive transcription, one would expect that highly active transcription sites are more likely to associate with large/bright clusters. Alternatively, there might be an optimal clustering regime for transcription response, which would be a novel result. This experiment would help address whether large or smaller clusters are driving the observed effect and/or whether large clusters are just passengers.

We thank the referee for raising an important concern regarding the functional role of detected TF clusters. We note that this concern was similarly raised by referee 1, which we have provided a comprehensive response with new experiments and analyses (please refer to page 7-9 of this letter). Below we first briefly summarize the newly obtained major conclusions regarding the role of TF clusters, address the question regarding low overlap rate, and then present analyses regarding the size of TF clusters.

- New results have provided much stronger supports for the functional roles of TF clusters at the levels of locus-specific binding and locus-specific transcriptional activation, and have thus led to a significant revision of the manuscript. Briefly, we now first present new data from DNA FISH assay as the evidence supporting the locus-specific DNA binding ability of TF clusters (**Line 233-251** in the main text, **Fig. 3b-c, Supplementary Fig. 2**). We then present new data from the two-color time-lapse imaging experiment as the evidence supporting the locus-specific transcriptional activation capability of TF clusters (**Line 253-268** in the main text, **Fig. 3d-e, Supplementary Fig. 3a-c, Supplementary Movie 1**). We next use the original snapshot data as the evidence supporting the DNA-binding dependent, locus-specific transcriptional activation of target genes by TF clusters (**Line 270-290** in the main text). Overall, these results are consistent with a picture that some TF clusters can indeed bind to the reporter loci, and activate the transcription of reporter genes.
- Regarding the relatively low overlap rate in the original snapshot data (i.e., 4% when measured using TF clusters as denominator, or 10% when using PCP foci as denominator), our new experiments and analyses provide at least two reasons. First, from the two-color temporal traces we found that a fraction of PCP-mCherry transcription signals did not have visible TF clusters during their lifetime, potentially

because of TF-cluster-independent transcription activation (or we did not catch the TF cluster). Second, even if PCP-mCherry signals were (apparently) activated with TF clusters, co-localization between PCP and TF signals could be detected for only about 10% the time during the lifetime of the transcription site (estimated based on the two-color traces). Thus, the low detected overlap ratio reported in the original manuscript arose from scenarios that could be experimentally accounted for. It should be noted that these snapshot data are now used as the evidence supporting that DNA binding capability is needed for locus-specific transcriptional activation. These discussions have been included in the revised text (**Line 294-301** in the main text).

- Regarding whether and how the size of detected TF clusters could play a role, we performed two analyses to investigate this issue. First, alongside with the referee suggestion, we used the snapshot data to examine the potential relationship between the intensity of the TF cluster (a close proxy of size) and the intensity of the PCP signal. We found an intriguing negative correlation between these two variables, indicating that smaller TF clusters could elicit stronger transcriptional responses (**Supplementary Fig. 3h**). Second, we used the new two-color images from the DNA FISH assay to compare the intensities of TF clusters that overlap with reporter gene loci with mean intensity of the three largest non-colocalizing TF clusters. We found that the former ones have significantly lower intensity (**Supplementary Fig. 2d**). Together, these two analyses are consistent with the picture that larger TF clusters do not possess stronger transcriptional activation capacities, which is an intriguing finding as the referee pointed out. Yet, it is not true that the large clusters are just “passengers” as the referee stated, because we could find examples where large clusters can also overlap with transcription sites in the two-color movie (**Supplementary Movie 1**). These results have been described in the revised text (**Line 245-248** and **Line 286-290** in the main text) and are also discussed in the discussion section (**Line 480-483** in the main text).

In sum, we believe these new results have provided key new insights into the locus-specific DNA binding and locus-specific transcriptional activation roles of TF clusters, and have thus significantly improved the manuscript.

2) In figure 4, the change in K_d (multiple orders of magnitude) seems the major driver of the effect compared to a more modest contribution from changes in the Hill coefficient. This raises the question whether clustering drives ultrasensitivity as claimed by the authors, or rather modulates the sensitivity range of the system via K_d changes. The robustness of the Hill coefficient result is also not totally clear, since the lower rapamycin concentrations only sample a small fraction of the Hill curve and therefore the Hill coefficient estimates could be biased for some conditions: at 0.1 nM rapamycin, TF concentrations seem to reach at most ~30,000 AU, almost an order of magnitude below the fitted K_d of ~200,000 AU. The authors should fit all the K_d on their curves while keeping the Hill coefficient fixed (e.g. using an average of the K_d s from their current fits), and then compare the fit quality of the fixed Hill coefficient results with

their current fit results; they should use a statistical metric such as the Akaike information criterion to decide whether the increase in goodness of fit justifies adding an extra parameter. If it turns out the clustering modulates the sensitivity range of the system via K_d changes (which is also interesting!), the text should be rewritten accordingly, e.g. claims that "the nonlinearity in the system becomes larger when the TF clustering propensity increases" should be modified.

We thank the referee for raising an important issue regarding the fitting of dose response curves. We have carried out a series of new analyses to address this issue, which provide supports for our original claim that Hill coefficient indeed varies across different rapamycin conditions. More specifically,

- Regarding the comment that "*the lower rapamycin concentrations only sample a small fraction of the Hill curve*", we agree that there could be a sampling bias that could affect the fitting. To address this, we compared the local sensitivities of response curves without fitting Hill functions. More specifically, we calculated the local sensitivity based on the slope of the data in the log-log space, and compared between two rapamycin concentrations. Note that the calculation of local sensitivity avoids potential complications with nonlinear fitting of Hill function. By doing so, we found that the response curve at high rapamycin concentration has a much larger slope compared to the one at low rapamycin concentration (**Supplementary Fig. 5h** and **Supplementary Fig. 6e**). This result is consistent with the comparison of Hill coefficient (i.e., global sensitivity) obtained from Hill function fitting. More importantly, this result indicates Hill function fitting was able to accurately capture the change in response sensitivity from the input-output data despite the potential sampling bias. This result has been described in the main text (**Line 347-355**).
- In addition to the analysis of local sensitivity, we carried out detailed analyses of Hill function fitting.
 - We first plotted the squared norm of the residual (resnorm) of the data fitted to a Hill function by fixing Hill coefficient at a range of values (from 1 to 6, **Supplementary Fig. 5g**). We found that the resnorm changes non-monotonically as a convex curve for most rapamycin conditions. Importantly, the minimum resnorm was obtained at different Hill coefficient values for different rapamycin conditions, with the Hill coefficient (corresponding to minimum resnorm) generally increases as rapamycin concentration increases. Note that for the very high rapamycin conditions, the Hill coefficient appears to drop slightly. This result is consistent with our claim that the clustering of TFs appears to modulate the Hill coefficient (i.e., the global sensitivity of the response).
 - We next used AIC to evaluate the fitting results by fixing Hill coefficient at different values (from 1 to 6). The results (**Figure R1**) show that similar to the previous results, the minimum AIC was obtained at different Hill coefficient values for different rapamycin conditions, and the Hill coefficient at the minimum AIC also increases as rapamycin increases, which however decreases after 1nM rapamycin. The overall trend is similar to the evaluation using resnorm.

Thus, a model with a varying Hill coefficient is more suitable for explaining the measured data.

Figure R1. AIC versus Hill coefficient for Hill function fitting of the input-output data of U2OS-7TetO at indicated rapamycin concentrations. Note that Hill coefficient was fixed for each fitting and AIC was calculated for the fitted model.

Figure R2. Evaluating model fitting using AIC or squared residual norm. In the fixed model, the Hill coefficient was fixed at 2.29, while in the unfixed model Hill coefficient was allowed to be fitted. AIC or resnorm was calculated with the model obtained by fitting the Hill function using the input-output data of U2OS-7TetO at indicated rapamycin concentrations.

- Following the referee’s suggestion, we further compared curve fitting with fixed vs. unfixed Hill coefficient based on AIC and resnorm measures (**Figure R2**). For AIC, we found that the two fitting methods (i.e., fixed vs. unfixed), the unfixed method is better compared to the fixed method when rapamycin is at low (0.05 and 0.1nM) or intermediate (1nM) level (**Figure R2** left). Within this range, Hill coefficient has doubled, indicating that fitting with unfixed Hill coefficient is preferred. Yet for higher rapamycin concentrations, the Hill coefficient in the unfixed method does not change as much, and the AIC score indicates that the fixed method is better. For resnorm measure, fitting using unfixed Hill coefficient consistently outperforms fitting using fixed Hill coefficient method (**Figure R2** right). It should be noted that AIC evaluates the predictive power of the model while resnorm evaluates the explanatory power of the model, and in our study, we aimed to construct a model to better explain the data in order to identify trends in the data (i.e., whether Hill coefficient changes). Thus, we think the model with unfixed Hill coefficient can much better explain the data and was able identify trend in data collected under different rapamycin concentrations.

Together, these results suggest that Hill coefficient (i.e., global sensitivity) is indeed

changing under different rapamycin conditions, and that an increase in TF clustering propensity can result in an increased Hill coefficient. We have now explicitly described and discussed the issue on Hill fitting (**Line 347-355** in the main text).

3) In the memory experiment, rapamycin-specific effects on translation and protein degradation could impact iRFP readout by modulating iRFP translation or stability. The authors should control that transcription activity tracks with the observed trend at the protein level, either using their PCP-based transcription reporter, or by RT-PCR.

We thank the referee for suggesting an important experiment to test whether the memory indeed occurs at the level of transcription (and not at the level of protein translation or stability). We have carried out new memory experiments to quantify the change in target gene's mRNA level over the same experimental time course illustrated in Fig. 5a. More specifically, we used two pairs of primers to quantify the expression of iRFP (i.e., the target gene) using qPCR over 50 hours post stimulation. Reassuringly, we found that at high clustering propensity the mRNA level decays much slower compared to the no rapamycin condition (**Supplementary Fig. 7d**), suggesting that the memory indeed occurs at the level of gene transcription instead of protein translation or stability. This result has been described in the revised text (**Line 413-417** in the main text). We believe that this new result has much strengthened our conclusion.

Minor points:

- Figure 3C/S2B: are there fewer clusters in the cells when there is no Dox which would skew the results? It would be helpful to include a "pseudo clusters" control to the no Dox condition. The expectation would be that there would be no enrichment in clusters in the no Dox above no Dox pseudo clusters.

We note that the total number of TF clusters in the no Dox condition is lower but comparable to the Dox condition (2500 vs. 3931 TF clusters). Following the referee's suggestion, we have included new analysis using "pseudo clusters" for the no Dox condition (**Fig. 3g**), and the result showed that the overlap fraction of pseudo clusters is comparable to the real TF clusters for the no Dox condition, which is in line with the referee's expectation that "*there would be no enrichment in clusters in the no Dox above no Dox pseudo clusters*".

- Figure 4b: axes are unclear.

We apologize for the lack of clarity. We have now relabeled the axes for Fig. 4b and other related figures.

- Error bars are needed in Figure 1 d-f, 4 c-e, some supplementary figures

We have now added error bars to the indicated figures and several other supplementary figures.

- Figure 5: c-d: n = ?

We note that data in Fig. 5c-d were from three biological replicates, which we have now indicated in the legend (**Line 1051-1052** in the main text). The rationale for combining data from the replicates was to show that all data exhibited a similar trend. Note that the distributions of three individual replicates behave similarly.

- initiation time is a term that could be confusing in the context of transcription as it often refers to the initiation step of each transcription cycle - “time to first burst” might be a better term to describe the induction of the system?

We thank the referee for suggesting an alternative term for describing the initiation of transcription. We agree that it is more accurate to describe it as “time to the first transcriptional burst”, which we have now incorporated in the revised text (**Line 309-310** in the main text and the related figure legend).

- In the memory experiment, could the author quantify the visible clusters remaining over time?

Following the reviewer suggestion, we carried out the suggested experiment to monitor the visible TF clusters in the memory experiment. While we performed imaging experiments on the CHO cell line in Supplementary Fig. 7e-f, these cells were imaged with only one single z-plane. In the new experiments, we imaged at 5 z-slices. More specifically, after a 12-hour pulse of doxycycline stimulation, we removed the doxycycline (while keeping rapamycin, as illustrated in Fig. 5b), performed z-stack imaging every 1 hour for ~2 days, and quantified the number of visible clusters during the memory experiment. Because we could not track single cells over such a long time course, we showed per-field-of-view quantification of TF cluster numbers over a 40 hour time course. By doing so, we found that the visible TF cluster number decreases gradually over the time course (**Supplementary Fig. 7h**), and that at later time points some cells do not have visible TF clusters. This result is consistent with the picture that TF clustering affected by the concentration of the TF.

- Could the authors quantify the lifetime of the clusters upon puromycin wash-off? This could narrow down models explaining the memory observed, e.g. do clusters remain physically present or do they confer long-lived transcription enhancement beyond their lifetime.

We note that in the memory experiment, we did not wash off rapamycin, i.e., the rapamycin was kept in the media and only doxycycline was washed off. As for the model of memory, we

think that our explanation based on affinity (i.e., Supplementary Fig. 9) is simple and convincing. This is because that we have a reliable measurement of K_d 's at different rapamycin concentrations for the CHO-Gal4 cell line (i.e., Supplementary Fig. 6a), and an increase in affinity at high rapamycin would theoretically lead to a sustained transcriptional response, which we validated with simulations.

Regarding the model suggested by the referee, i.e., TF clusters could elicit sustained transcriptional response at the level of individual gene loci. While this model does sound appealing, our new two-color imaging results suggested that this model does not agree with the observed behaviors of individual gene loci (**Fig. 3d-e, Supplementary Fig. 3a-c, Supplementary Movie 1**). More specifically, in the two-color traces, we found that TF clusters and nascent transcription sites could interact in different ways (**Supplementary Fig. 3c, Supplementary Movie 1**). In some scenarios, TF clusters exhibited frequent interactions with the nascent transcription sites, and there was only a small fraction of scenarios that TF clusters elicited sustained transcriptional response.

Therefore, we think the affinity explanation for memory-like behavior would be more plausible, and in light of our new two-color experiments this explanation could arise from the scenario that TF clusters can more quickly and easily bind to the target loci compared to non-clustered TFs.

- The authors should clearly define “ultrasensitivity” in the introduction.

We thank the referee for the suggestion. We have now clearly defined ultrasensitivity, among other terminologies used in the text, in the introduction (**Line 56-58** in the main text).

- Page 4, second last paragraph - “there are additional parameters such as protein expression level that can affect the clustering propensity at the level of individual cells” - do the authors observe any evidence of this in their system?

In our original manuscript, Supplementary Fig. 1b-c included plots that indicated the concentration of TFs and TF cluster properties of individual cells. These results indicated that TF concentration appear to after cluster number and diameter. In our new CHO memory experiment described above, the results also provided evidence that TF concentration can play a role in the formation of visible TF clusters (**Supplementary Fig. 7h**).

- Page 6, second last paragraph - the authors speculate the short dwell time of TF clusters on DNA being one of the reasons for the low % of overlaps between induced clusters and the active transcription sites. Do any of the observed clusters move around over time, and can they measure the dwell time of clusters at transcription sites using their PCP reporter?

As described in the previous responses, we have performed new two-color imaging experiments using U2OS-7TetO cell line to examine the interaction between TF clusters and

nascent transcription sites. Using these new two-color traces, we have showed that TF clusters do indeed bind on/off at the transcription sites. We can also quantify the dwell time of TF clusters binding to the transcription sites, which is exponential like (**Supplementary Fig. 3b**). These results suggest that our original speculation was correct in that the generally short TF cluster dwell times contribute to the low overlap between TF clusters and nascent transcription sites.

Reviewers' Comments:

Reviewer #2:

Remarks to the Author:

I appreciate the authors' effort to add extra controls to solidify their experimental observations. While the data stands as is, I have concerns about the authors' current interpretation and writing of the manuscript which is focused on minor aspects while ignoring the main features of the phenomena observed:

- This goes for the changes in Hill coefficient (1 to 2.5) which remain minor drivers of the observed effect compared to the ~ 3 orders of magnitude difference in Kd observed in response to rapamycin addition. The overwhelming effect of rapamycin on the response of the system is thus a large increase in sensitivity, not the emergence of ultrasensitivity (Fig. 4 b-d). In order to assert that ultrasensitivity is the main driver of the response, the authors should use TetR mutants with higher KDs at high rapamycin concentrations in order to maintain similar KDs across the entire rapamycin response curve. In the absence of such a control for KDs, the paper should be rewritten to de-emphasize the ultrasensitive aspect and change the title, as the focus on the ultrasensitivity is misleading from the main features of the system.

Similarly, for bimodality: the apparent emergence of bimodality is a trivial result of the change in KD (Supp figure 5), and likely does not correspond to a fundamental change in the behavior of the system. Because the KD is so high at low [rapamycin], the system never explores concentrations above the KD; even if the system was ultrasensitive in those low rapamycin conditions, we would be unable to observe its bimodality because the input is not appropriately sampled. Therefore the results are inconclusive regarding the source of the bimodality.

I recommend to substantially change the writing to take into account those major limitations of the system.

Dear Reviewer:

Thank you very much for the critical and very helpful comments on our manuscript. Based on your comments and suggestions, we have made significant revisions to the manuscript to address all the concerns raised. The key revisions are summarized below and more detailed descriptions are included in the point-by-point reply.

- We now describe that TF clustering modulates both parameters (K_d and Hill coefficient) of the gene regulation function (i.e., input-output function), instead of focusing on the modulation of Hill coefficient as in the original manuscript. Importantly, we emphasize that the observation that K_d is modulated by a much larger extent compared to Hill coefficient. As such, many modifications were made in the revision, including the manuscript title, abstract, main text, and figure legends.
- In addition to accurately describing how parameters of the gene regulation function are modulated by TF clustering, we now clearly explain how such parameter modulations (i.e., the modulations of K_d and Hill coefficient) contribute to bimodality and sustained response behavior, with the help of new simulations. These mechanistic clarifications are important, as they help to delineate the roles of parameters in the emergent behaviors.
- We have thoroughly discussed the limitations of the study as well as the unexplained observations, including the limitation of imaging resolution for co-localization analysis, the differential modulations of K_d and Hill coefficient by TF clustering, and the relative contributions of the modulations of two parameters to bimodality.

We greatly appreciate the opportunity to clarify several key issues that you raised. With all these revisions, our manuscript now presents a clear and comprehensive dissection and delineation of the roles of TF clustering in gene regulation, which has significantly advanced the current knowledge on transcription protein clusters and condensates.

Below please find a summary of updates to figures and figure titles:

- Figure 4: Figure title was changed to “TF clustering propensity modulates the gene regulation functions of synthetic TFs”, which better describes the data and avoids the focus on ultrasensitivity.
- Figure 6: Figure title was updated in order to better summarize the main findings.
- Supplementary Fig. 5: panels e and f were switched in order to be consistent with the text edits; green boxes were added to panels c, e, and f in order to label the region of interest; figure title was slightly modified.
- Supplementary Fig. 6: panels b and c were switched, as well as panels g and h, in order to be consistent with the text edits; figure title was slightly modified.
- Supplementary Fig. 9: panels e and f were added to include new simulation results that clarify the roles of the two parameters in memory-like behavior.

Point-by-point reply

"- Figure 3; The authors show 4% of TF clusters are colocalized with PCP foci at 100 nM rapamycin with doxycycline. According to Figure 1e with 100 nM rapamycin, 5 TF clusters on average are in a single cell. It means that only one cell among five cells has one colocalization. 7 The authors argued that they show the direct evidence of TF clustering effect on transcription, but it is hard for me to agree that the 50-fold change in reporter transcription was dominated by this one colocalization among five cells. Of course, as the authors pointed out, clustering might occur but not be captured by microscope. I agree with this statement, but I suggest detailed analysis to address it. Firstly, how much percent are the PCP foci colocalized with TF clusters? (# of colocalization / total # of PCP foci) Also, to confirm the locations and the number of reporter genes, DNA-FISH for PP7 is recommended. Then the authors can analyze the colocalizations of TF clusters and the reporter gene loci.

It helps to understand how many reporter genes are on activation during imaging time and the dynamics between TF clusters and reporter genes."

Reviewer #2 suggests either a super-resolution analysis to understand the dynamics of TF cluster foci and reporter gene activation or toning down the claims of the manuscript to account for this limitation of the study.

We thank the reviewer for pointing out the shortcoming of some of the imaging data. We agree that the co-localization analysis could suffer from the limitation in imaging resolution and may thus fail to distinguish between co-localization events with or without physical interactions. Despite such limitations, these results still provide evidence consistent with the direct activation of reporter transcription by TF clusters.

We have now clearly pointed out and discussed the limitations, and have toned down related claims.

- In the result section related to co-localization analysis, we have now pointed out the limitation related to imaging resolution a couple of times. Line 271-274: "*It should be noted that these spatiotemporal analyses of interactions relied on the visual inspection of co-localization between two fluorescent signals from two different channels, which may not be accurate enough due to the diffraction-limited resolution of our microscope*"; Line 301-305: "*Of note, these results likely also suffer from the same limitation as noted above for the spatiotemporal analysis, i.e., light diffraction in our microscope precludes the analysis of spatial co-localization at a high enough resolution. As such, the co-localization between two signals indicates the spatial proximity between two biological entities, and may not suggest their actual interactions.*"
- In the discussion section, we have included new sentences to clarify such limitations. Line 523-526: "*Of note, our co-localization analysis in both the snapshot data and the temporal data could suffer from limitations arising from limited spatial resolution and may thus fail to distinguish between co-localization events with or without physical interactions.*".

- Related claims have been toned down. For example, we replaced “key evidence” with “evidence”, replaced “suggest” to “implicate”, added “appear to” in front of “display binding and unbinding dynamics”, etc.

We believe that these modifications have largely toned down the claims related to static and dynamic co-localization analysis, and have appropriately alerted the readers on the potential limitations of our analysis.

Reviewer #2 (Remarks to the Author):

I appreciate the authors' effort to add extra controls to solidify their experimental observations. While the data stands as is, I have concerns about the authors' current interpretation and writing of the manuscript which is focused on minor aspects while ignoring the main features of the phenomena observed:

- This goes for the changes in Hill coefficient (1 to 2.5) which remain minor drivers of the observed effect compared to the ~3 orders of magnitude difference in K_d observed in response to rapamycin addition. The overwhelming effect of rapamycin on the response of the system is thus a large increase in sensitivity, not the emergence of ultrasensitivity (Fig. 4 b-d). In order to assert that ultrasensitivity is the main driver of the response, the authors should use TetR mutants with higher K_D s at high rapamycin concentrations in order to maintain similar K_D s across the entire rapamycin response curve. In the absence of such a control for K_D s, the paper should be rewritten to de-emphasize the ultrasensitive aspect and change the title, as the focus on the ultrasensitivity is misleading from the main features of the system.

We greatly appreciate the important concerns raised by the reviewer regarding the interpretations of certain data and the focus of the features of our system. We agree that certain data were not clearly and appropriately interpreted, and that the focus of the original manuscript was too much on ultrasensitivity. To address these concerns, we have performed additional simulations to clarify mechanistic interpretations and have substantially re-written the manuscript to ensure accurate descriptions and interpretations of the data. Additionally, we have also thoroughly discussed potential limitations. More specifically:

- To clearly and accurately interpret the roles of TF clustering from the data, we now describe that TF clustering modulates both parameters (K_d and Hill coefficient) of the gene regulation function (i.e., input-output function), instead of focusing on the modulation of ultrasensitivity. Because of this, many changes have thus been made:
 - The original title of the manuscript (“*Ultrasensitivity and sustained transcriptional response with chemically controlled transcription factor clustering*”) is now replaced by “*Modulating gene regulation function by chemically controlled*”

transcription factor clustering". We believe that this new title accurately describes the main findings in an unbiased manner, i.e., TF clustering modulates parameters of the gene regulation function, which then leads to emergent behaviors including bimodality and memory-like behavior.

- The abstract of the manuscript has also been updated to describe the modulation of the gene regulation function as a whole, instead of focusing on the modulation of ultrasensitivity. Importantly, following the suggestion of the reviewer, we now highlight in the abstract that "*TF clustering propensity modulates gene regulation function by significantly tuning the effective TF binding affinity and to a lesser extent the ultrasensitivity*". We believe that this is an accurate summary of the data because TF clustering has a stronger influence on the K_d compared to the Hill coefficient, as pointed out by the reviewer.
- Many modifications to the main text and figure titles/legends have been made to in order to consistently explain the messages carried in the updated title and the updated abstract. As examples:
 - a) In the introduction, we now write "*it remains to be further determined whether and how clustering (or condensation) represents a general control parameter for the quantitative modulation of the gene regulation function, i.e., the functional dependency of target gene output on the concentration of the TF*" (Line 70-73). And in the summary paragraph, we now summarize how gene regulation function is modulated and the resulting emergent behaviors (Line 90-102).
 - b) In the results, we have heavily modified the section related to Fig. 4 and changed the section title to "High TF clustering propensity enables bimodal target responses". In this section, we now clearly describe that "*both the effective binding affinity and the ultrasensitivity of the gene regulation function are modulated by TF clustering propensity, and the effective binding affinity appears to be modulated by a much larger extent compared to the ultrasensitivity*" (Line 370-374). This is in line with the suggestion of the reviewer and greatly improves the accuracy of the interpretation.
 - c) In the discussion part, we now clearly discuss the finding that "*TF clustering confers a stronger modulation on the effective TF binding affinity than on the Hill coefficient for all three systems characterized*" (Line 532-534). We also speculated a potential explanation (Line 534-538).
 - d) Many more changes are highlighted in the track change version of the revised manuscript.
- In addition to accurately describing how parameters of the gene regulation function are modulated by TF clustering, we now clearly explain how such parameter modulations contribute to bimodality and sustained response behavior. These mechanistic clarifications are important, as they help to delineate the roles of parameters (i.e., K_d or Hill coefficient) in the emergent behaviors. Specifically:
 - For bimodality, we have now clearly described that three factors play a role in conferring bimodality: 1) the range of input signals; 2) the large shift in K_d ; 3) and

the increase in ultrasensitivity. We also state that the first two factors are obvious and that the last factor (ultrasensitivity) is not as obvious: *“It is apparent that in addition to the range of input signals, the decreasing K_d value as rapamycin increases is important for the appearance of bimodality. Yet, the contribution from the increase in Hill coefficient is less obvious”* (Line 377-380). We then used the data to show that Hill coefficient likely synergizes with K_d to confer bimodality: *“We observed that among these conditions, the bimodality index increases when Hill coefficient increases (Supplementary Fig. 5c), indicating that the modulation of ultrasensitivity likely contributes to the appearance of bimodality, together with the modulation of the effective binding affinity”* (Line 382-385).

- For sustained response (i.e., memory-like) behavior, we originally showed simulation results using measured gene regulation functions at low and high rapamycin conditions, which recapitulated the experimental observation. We have now performed additional simulations to further clarify the role of the two parameters (i.e., K_d and Hill coefficient). More specifically, we performed two additional sets of simulations by fixing either K_d or Hill coefficient. We found that the modulation in K_d is essential for conferring memory-like behavior, while the modulation of ultrasensitivity does not contribute to this behavior (**Supplementary Figs. 9e-f**, Line 482-489).
- Moreover, we have thoroughly discussed the unexplained observations in our system.
 - We pointed out in the discussion section that the differential modulations of K_d and Hill coefficient by TF clustering is intriguing and provided speculations for why the modulation of K_d is large (Line 532-538). We also raise the question of whether the current observation represents a general functional principle of TF clustering (Line 540-544).
 - We clarified in the discussion section that the relative contributions of the modulations of two parameters to bimodality remain undetermined (Line 553-554).

We believe that these modifications have significantly improved the interpretation of the data and have much clarified the roles of TF clustering in the modulation of gene regulation function. Thus, the clarity of the manuscript has been greatly enhanced and the concerns raised by the reviewer should have been addressed.

Similarly, for bimodality: the apparent emergence of bimodality is a trivial result of the change in KD (Supp figure 5), and likely does not correspond to a fundamental change in the behavior of the system. Because the KD is so high at low [rapamycin], the system never explores concentrations above the KD ; even if the system was ultrasensitive in those low rapamycin conditions, we would be unable to observe its bimodality it because the input is not appropriately sampled. Therefore the results are inconclusive regarding the source of the bimodality.

I recommend to substantially change the writing to take into account those major limitations of the system.

We greatly appreciate the reviewer for raising an important issue regarding the mechanism for bimodality. This comment is along the line of the previous one and concerns about the mechanistic interpretation of bimodality.

As in the response to the point above, we now clearly explain how parameter modulations contribute to bimodality, in which we describe three factors that play roles (please see above). In addition to this new explanation in the results section, we also include a new paragraph in the discussion section, where we discuss in detail about the “trivial” and “non-trivial” aspects of our findings on bimodality. More specifically, this discussion reads: “*While the emergence of bimodality appears to be a trivial consequence resulting from the modulation of the effective TF binding affinity, the fact that TF clustering can significantly tune the effective binding affinity is rather unexpected, which could provide insights into the many bimodally distributed genes in the transcriptome*” (Line 548-551).

Thus, while we agree with the reviewer that the emergence of bimodality seems trivial and intuitive, we think it is nontrivial that we were able to pinpoint the detailed mechanism, which largely arises from the modulation of K_d by TF clustering (when inputs are appropriately sampled). Moreover, this result highlights the advantage of our synthetic system, as we clearly demonstrated that a single parameter tuning of the TF (i.e., the clustering propensity) can result in a drastic change in the output, which could help to clarify mechanisms of bimodality for many natural genes.

Overall, we greatly appreciate the opportunity to clarify several key issues raised here by the reviewer. With all these newly added revisions, our manuscript presents a clear and comprehensive dissection and delineation of the roles of TF clustering in gene regulation, which have significantly advanced the current knowledge on TF clusters and condensates.

Reviewers' Comments:

Reviewer #2:

Remarks to the Author:

All my concerns have been addressed - I especially appreciate the additional simulations that clarify the respective contributions of K_d and ultrasensitivity to bimodality and memory. These have largely improved the mechanistic interpretation of the authors experiments.

Point-by-point reply

Reviewer #2 (Remarks to the Author):

All my concerns have been addressed - I especially appreciate the additional simulations that clarify the respective contributions of K_d and ultrasensitivity to bimodality and memory. These have largely improved the mechanistic interpretation of the authors experiments.

We thank the reviewer for the very positive response. We greatly appreciate all the previous comments and suggestions, which have helped us to significantly improve the manuscript.